# CoDoL: Conditional Domain Prompt Learning for Out-of-Distribution Generalization

**Min Zhang**                                       *mzhang@cs.ecnu.edu.cn*
*East China Normal University*

**Yuyin Wang**[†]                                    *yuyinws@gmail.com*
*Xidian University*   [†]*Corresponding authors*
*Zhejiang Key Laboratory of Artificial Intelligence of Things (AIoT) Network and Data Security*

**Zhongxiang Dai**                                   *daizhongxiang@cuhk.edu.cn*
*The Chinese University of Hong Kong, Shenzhen*

**Zhikang Chen**                                     *zhikang.chen@eng.ox.ac.uk*
*The University of Oxford*

**Jie Zhou**                                         *jzhou@cs.ecnu.edu.cn*
*East China Normal University*

**Miao Liu**                                         *miaoliu@mail.tsinghua.edu.cn*
*Tsinghua University*

**Sen Cui**[†]                                       *cuis@mail.tsinghua.edu.cn*
*Tsinghua University*   [†]*Corresponding authors*

**Reviewed on OpenReview:** *https://openreview.net/forum?id=MDxhbeE21D*

## Abstract

Recent advances in pre-training vision-language models (VLMs), *e.g.*, contrastive language-image pre-training (CLIP) methods, have shown great potential in learning out-of-distribution (OOD) representations. Despite showing competitive performance, the prompt-based CLIP methods still suffer from: i) *inaccurate text descriptions*, which leads to degraded accuracy and robustness, and poses a challenge for zero-shot CLIP methods. ii) *limited vision-language embedding alignment*, which is one important factor affecting generalization performance. To tackle the above issues, this paper proposes a novel Conditional Domain prompt Learning (CoDoL) method, which utilizes readily-available *domain information* to form prompts and contributes to improved vision-language embedding alignment, which we identify as one factor underlying the observed OOD generalization gains. To capture both instance-specific and domain-specific information, we further propose a lightweight Domain Meta Network (DMN) to generate input-conditional tokens for images in each domain. Extensive experiments on four OOD benchmarks (PACS, VLCS, OfficeHome, and DigitDG) validate the effectiveness of our proposed CoDoL method in terms of empirically improves vision-language embedding alignment across four DG benchmarks, which we present as a contributing factor (rather than the sole cause) of the observed OOD gains.

## 1 Introduction

Deep learning methods generally rely on the independent and identically distributed (IID) assumption that the distributions of training data and testing data are the same (Gänssler & Stute, 1979; He et al., 2015; 2016a;b). However, in real-world applications, out-of-distribution (OOD) generalization is a ubiquitous

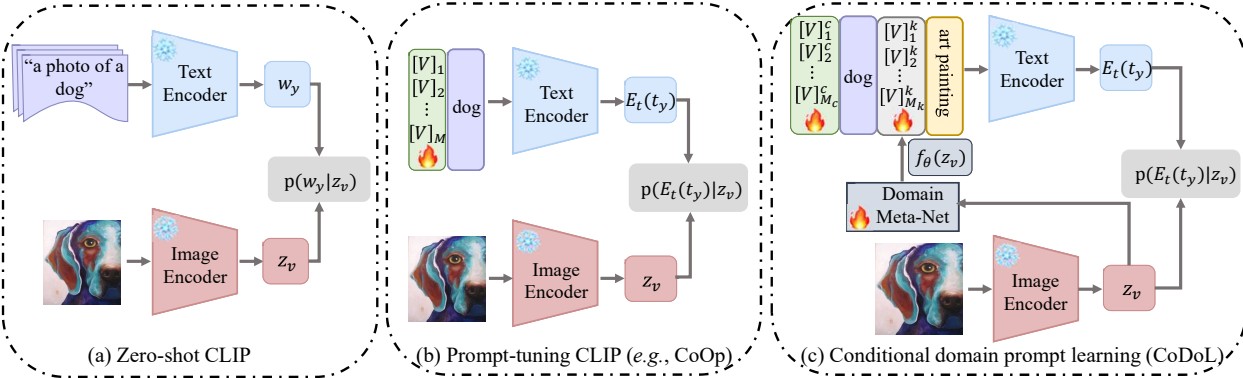

Figure 1: Comparison of the framework based on CLIP models. (a) Zero-shot CLIP uses the fixed context. (b) Prompt-tuning CLIP designs the learnable context. (c) CoDoL not only models the learnable context for the class prompt but also drafts the domain prompt to align the vision and language representations. Moreover, a lightweight domain meta-net is also proposed to generate the input-conditional domain-specific information. The snowflake and fire mean frozen and learnable, respectively.

problem, where the distribution of testing data differs from the training data, leading to the significant performance degradation (Zhang et al., 2022; Yao et al., 2023). Many methods have been proposed to solve the out-of-distribution generalization problem, including: (1) causal learning (Cha et al., 2022; Lin et al., 2022a; Arjovsky et al., 2019), adversarial learning (Ajakan et al., 2014; Li et al., 2018a) and meta learning (or named learning to learn) (Balaji et al., 2018; Khattak et al., 2022), and so on.

Previous studies used supervised pre-trained models and carefully-designed transfer learning algorithms for achieving OOD generalization (Gulrajani & Lopez-Paz, 2020; Kim et al., 2022). Recently, instead of learning from human-labeled data, vision-language pre-training models (VLMs) seek to learn from naturally formed supervision of web-scale image-language pairs (Radford et al., 2021a; Jia et al., 2021), which exhibits impressive zero-shot learning performance and outperforms models trained from only labeled images. The VLMs, such as the contrastive language-image pre-training model (CLIP) (Radford et al., 2021b), also empirically outperform traditional supervised learning methods on OOD downstream tasks in terms of zero-shot performance (see Tables 1 and 2 for more details), which reveals a promising research direction toward OOD generalization in real-world downstream tasks.

Specifically, as shown in Figure 1, CLIP methods first compute the similarity between images and embedded words for each category, then train the text and image encoders by maximizing the similarity. Therefore, the vision-language embedding alignment heavily affects the generalization performance of the standard CLIP methods. Similar conclusions also could be found in previous works (Shu et al., 2023; Menon & Vondrick, 2022; Goyal et al., 2022). Figure 1 (a) illustrates the zero-shot CLIP method, which uses a fixed text to test the performance of the pre-training encoder on downstream tasks. Figure 1 (b) illustrates the prompt-tuning CLIP methods, which use the learnable context and update only a small number of additional parameters during the fine-tuning phase, thereby reducing the training and storage burdens.

Despite the competitive performance in OOD downstream tasks, these methods still suffer from the following issues: i) *inaccurate text descriptions*, which leads to degraded accuracy and robustness, and poses a challenge for zero-shot CLIP methods (Zhou et al., 2022a;b). ii) *limited vision-language embedding alignment*, which serves as a key factor affecting the generalization performance of the prompt-tuning CLIP methods (Zhang et al., 2021; Li et al., 2022; Bose et al., 2023). The two core challenges significantly affect the generalization performance for pre-training vision-language models under the out-of-distribution setting.

To tackle the above issues, in this paper, we propose to utilize *domain information* to improve the alignment of the embedding spaces of CLIP image and text encoders, thus improving the OOD generalization performance. This domain information could be different measuring circumstances, locations, times, experimental conditions, external interventions, contexts, and so forth, which are readily available in practice (Arjovsky et al., 2019; Krueger et al., 2021; Lin et al., 2022b). Figure 2 illustrates a toy example of the benefits

of utilizing domain information for improving OOD generalization using the PACS benchmark dataset (Li et al., 2017), which contains four domains, *i.e.*, Photo, Art Painting, Cartoon and Sketch. Compared with the zero-shot CLIP method, we additionally introduce the domain token and obtain a novel prompt form, *i.e.*, "a photo of a `[CLASS] [DOMAIN]`", with the added domain vectors. The proposed zero-shot CLIP with domain information achieves 93.17% accuracy, whereas the previous zero-shot CLIP method only achieves 90.32% accuracy, validating the effectiveness of introducing the domain information for OOD generalization.

Given that the domain information can improve out-of-distribution (OOD) generalization, we propose Conditional Domain prompt Learning (CoDoL), a simple and effective parameter-efficient method to improve the OOD generalization of CLIP on downstream tasks. Specifically, CoDoL first models a class prompt's context words with learnable vectors following the previous works (Zhou et al., 2022a;b). Then, we design the learnable domain prompt's context vectors to capture the domain information in the prompt-tuning phase. Because OOD generalization aims to learn knowledge from multiple training domains and transfer it to unseen testing domains, in this paper, the proposed domain prompt is domain-agnostic parameters that are used to learn domain-invariant representation and generalize to unseen testing domains. Furthermore, to capture instance-specific information, we design a lightweight neural network named Domain Meta Network (DMN) to generate input-conditional tokens for images in each domain, which is then concatenated with the learnable domain-specific context vectors. Extensive experiments are conducted on four different OOD benchmarks, and experiments demonstrate the state-of-the-art performance of CoDoL compared with ImageNet pre-trained and CLIP pre-trained baselines. Our main contributions are summarized as follows:

- We propose a prompt-tuning CLIP method (CoDoL), which utilizes readily-available *domain information* to form prompts and improves the vision-language embedding alignment for improving OOD generalization.

- We further propose a lightweight neural network (DMN) for capturing both instance-specific and domain-specific information, by generating input-conditional tokens for images in each domain, and then concatenating them with the learnable domain-specific context vectors.

- We conduct extensive experiments on four OOD benchmark datasets with various backbones, showing that CoDoL can improve the alignment of the embedding spaces of CLIP image and text encoders, thus improving the OOD generalization performance.

## 2 Related Work

### 2.1 Out-of-Distribution (OOD) Generalization

OOD generalization aims to train a robust model from one or multiple training domains and make it predict well on previously unseen testing domains. A variety of OOD strategies have been proposed to overcome distribution shifts, including causal learning (Arjovsky et al., 2019; Sagawa et al., 2019; Lin et al., 2022b), meta learning (Balaji et al., 2018; Li et al., 2018a; Shu et al., 2021; Volpi et al., 2021), adversarial learning (Ganin et al., 2016; Li et al., 2018b; Sicilia et al., 2021), and so on. These methods study used supervised pre-trained models and have achieved significant success on real-world OOD shifts (Gulrajani & Lopez-Paz, 2020; Kim et al., 2022). Recently, with the rise of such powerful vision-language models (VLMs) like the contrastive language-image pre-training model (CLIP) (Radford et al., 2021b), VLMs strive to learn from naturally formed supervision of web-scale image-language pairs rather than learning from human-labeled data. This paradigm enables to learn from diverse domains and recognizing concepts from real-world applications (Radford et al., 2021a; Jia et al., 2021). Based on the pre-trained model by CLIP, some works (Zhang et al., 2021; Cha et al., 2022; Shu et al., 2023; Bose et al., 2023) have been proposed and achieved an impressive performance compared with the pre-trained model by ImageNet (see more details in Tables 1 and 2). This paper addresses the OOD problem based on the robustness of the CLIP model as well.

### 2.2 Vision-Language Models Pre-training

The workflow of pre-training and fine-tuning has become a popular paradigm for solving many downstream tasks in the NLP or computer vision field. Recently, inspired by the success of pre-training with web-scale

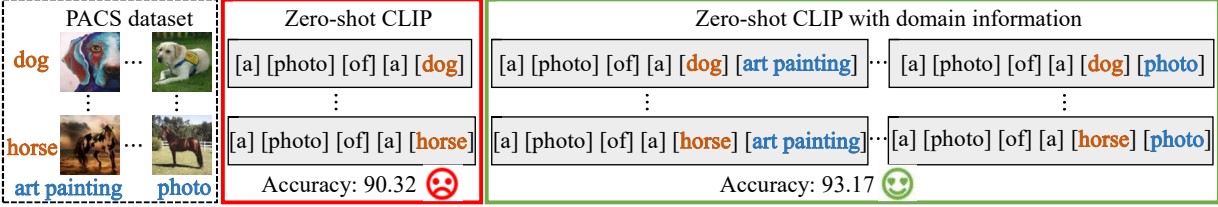

Figure 2: Motivation of CoDoL. Compared with zero-shot CLIP, we additionally introduce the domain information and obtain the new prompt of "a photo of a `[CLASS] [DOMAIN]`". The new prompt can help CLIP to better align the two embedding spaces, further improving the OOD generalization.

unlabeled data in the NLP field, Radford *et al* (Radford et al., 2021b) propose CLIP, which has a rich cross-modal representation and can solve a wide range of tasks without additional supervision. The main idea behind CLIP is to create a system that understands and interprets images and text descriptions in a similar way to humans. In the pre-training phase, CLIP uses the contrastive loss to learn the relationship between the image and text drawn from the web. In the inference phase, the pre-training vision-language model has a significant performance in various downstream tasks. Recently, various variant methods have also been proposed to solve the challenge problem under different fields, *e.g.*, image detection, Semantic segmentation, video processing, and so on (Gao et al., 2021; Zheng et al., 2022). In this paper, we focus on using CLIP as the pre-training visual-language model to improve performance and robustness on downstream tasks of the out-of-distribution (OOD) generalization.

## 2.3 Parameter-Efficient Transfer Learning

Parameter-Efficient Transfer Learning (PETL) updates only a small number of pre-trained or additional parameters during the fine-tuning phase of downstream tasks, which reduces the training and storage burdens. Representative prompt tuning optimizes learnable tokens inserted into the input token sequence while freezing the pre-training vision and language encoders (Radford et al., 2021b; Zhang et al., 2021; Zhou et al., 2022b;a; Khattak et al., 2022; Jia et al., 2022; Li et al., 2022; Zhu et al., 2022). Some works have been proposed to design different prompts to improve the performance in the distribution shift downstream task (Zheng et al., 2022; Ge et al., 2022; Bose et al., 2023). We compare the difference between previous prompt-tuning CLIP works and our CoDoL: (1) We aim to improve the performance of prompt-tuning CLIP in real-world OOD shifts instead of some variants of ImageNet (Russakovsky et al., 2015). (2) StyLIP (Bose et al., 2023) is the current SOTA prompt-tuning CLIP method in real-world OOD shifts. It uses the style features at different visual encoder levels to learn the individual prompt tokens and judicious exploration of multi-scale visual content features in the prompt learning phase. (3) CoCoOp (Zhou et al., 2022a) also uses a lightweight neural network to learn input-conditioned class information. It aims to improve the generalization of unseen classes in the testing phase, but our CoDoL learns input-conditioned domain representation and generalizes to unseen testing domains. Recent domain-aware prompt learning. A growing line of work explores domain-aware prompting for VLMs under distribution shift. Cheng et al. (2024) propose a Disentangled Prompt Representation that decouples domain-invariant and domain-specific prompts. Chi et al. (2024) introduce a Visual Domain Prompt Generation framework targeted at test-time distribution shift, generating prompts on the visual side. Zhao et al. (2024) learn a Domain-Invariant Prompt with explicit invariance regularization. Xu et al. (2024) ensemble disentangled domain-specific prompts to improve DG.

Our CoDoL differs from these works in three aspects: (i) we explicitly model an instance-conditional domain offset via the DMN, rather than only learning static domain-invariant/specific prompts; (ii) we operate purely on the text prompt side and require no test-time intervention; (iii) we marginalize over training-domain prompts at inference, so domain labels are not needed at test time.

## 3 Preliminary

In this section, we first introduce the problem setup and notations of out-of-distribution (OOD) generalization. Then, we provide brief reviews on prompt-based CLIP methods, including zero-shot CLIP (Radford

et al., 2021b) and prompt-tuning CLIP (Zhou et al., 2022b;a). Finally, we propose a novel conditional domain prompt learning (CoDoL) with a lightweight domain meta network (DMN), which respectively enhances the vision-language embedding alignment and captures the instance-specific information for improving the OOD generalization performance of the CLIP.

## 3.1 Problem Setup and Notation

We consider the setting where one predicts the label $y \in \mathcal{Y}$ based on the input image $x \in \mathcal{X}$. Besides, we assume that each sample is associated with a domain label $k \in \{1, 2, \ldots, K\}$, which could be different measuring circumstances, locations, times, experimental conditions, external interventions, contexts, and so forth, and are readily available in practice (Arjovsky et al., 2019; Krueger et al., 2021; Lin et al., 2022b). Let $\mathcal{D}_{tr} = \{(x_i, y_i, k_i)\}_{i=1}^{n}$ denotes a training dataset of $n$ samples associated with a training distribution $p_{tr}$. OOD generalization aims to train a classifier with an accuracy guarantee over the unseen test distribution $p_{ts}$ when distribution shift occurs, *i.e.*, $p_{ts} \neq p_{tr}$.

## 3.2 Prompt-Based CLIP Methods

Vision-language pre-trained models (VLMs) (*e.g.*, CLIP) have recently demonstrated great potential in learning generic visual representations and allowing zero-shot transfer to a variety of downstream classification tasks (Radford et al., 2021b;a; Jia et al., 2021). Next, we illustrate the learning diagram for pre-training CLIP and the inference paradigm for zero-shot CLIP and prompt-tuning CLIP.

**Pre-training CLIP.** The pre-training phase of CLIP (Radford et al., 2021b) uses an image encoder $E_v$ and a text encoder $E_t$. Specifically, the image encoder $E_v$ converts a given image $x \in \mathbb{R}^{3 \times W \times H}$ into a $D$-dimensional feature vector $z_v \in \mathbb{R}^D$, where $W$ and $H$ denote the width and height of the image, respectively. The text encoder $E_t$ converts a given sequence of word tokens to a vectorized representation $z_t \in \mathbb{R}^{L \times B}$, where $L$ represents the text length and $B$ represents the embedding dimension. In this paper, we let $E_v$ be either ResNet (He et al., 2016a) or ViT (Dosovitskiy et al., 2020), and $E_t$ be Transformer (Vaswani et al., 2017). CLIP adopts a contrastive loss to learn a joint embedding space from a large-scale dataset composed of paired images and captions, which maximizes for each image the cosine similarity with the matched text.

**Zero-Shot CLIP.** In the zero-shot transfer phase, as shown in Figure 1(a), CLIP tries to match an image with a textual description. Specifically, $E_v$ extracts the image features $z_v$ for each image $x$ and $E_t$ generates a set of weights $\{w_y\}_{y \in \mathcal{Y}}$ associated with class $y \in \mathcal{Y}$. Each weight $w_y$ is derived from a prompt that has the form of "a photo of a [CLASS]", where the class token is replaced by a specific class name, such as "dog", "horse", or "car", *etc*. The predicted class probability is then computed as follows:

$$p(y|x) = \frac{\exp(\cos(w_y, z_v)/\tau)}{\sum_{y \in \mathcal{Y}} \exp(\cos(w_y, z_v)/\tau)}, \tag{1}$$

where $\tau$ is a temperature parameter and $\cos(\cdot, \cdot)$ denotes the cosine similarity.

**Prompt-Tuning CLIP.** The prompt template is crucial for zero-shot CLIP (Radford et al., 2021b), design of which unfortunately requires expertise and heavy time.

Therefore, the tuning-based prompt template has been widely considered (Zhou et al., 2022a;b; Khattak et al., 2022). For example, as illustrated in Figure 1(b), the representative CoOp (Zhou et al., 2022b) models a class prompt as:

$$t = [V]_1, [V]_2, \cdots, [V]_M + [\text{CLASS}], \tag{2}$$

which consists of a set of context token vectors $\{[V]_m\}_{m=1}^{M}$ shared by all classes or specific to each class, and a class token [CLASS].

Then, by forwarding the class prompt to the text encoder, we can obtain the class probability of a given image feature $z_v$ as:

$$p(y|x) = \frac{\exp(\cos(E_t(t_y), z_v)/\tau)}{\sum_{y \in \mathcal{Y}} \exp(\cos(E_t(t_y), z_v)/\tau)}. \tag{3}$$

### 3.3 Conditional Domain Prompt Learning

To improve the OOD generalization performance, we propose a conditional domain prompt learning (CoDoL) method, which utilizes the readily available domain information to form prompts and improves the vision-language embedding alignment for prompt-based CLIP methods. Different from previous prompt-tuning methods, we not only model the learnable class context to construct the class prompt, but also design the trainable domain context to generate the domain prompt. The new prompt input to the text encoder $E_t$ is designed with the following form:

$$
\begin{aligned}
t &= \text{prompt} + \texttt{[CLASS]} + \texttt{[DOMAIN]}, \quad \text{and} \\
\text{prompt} &= \underbrace{[V]_1^c, [V]_2^c, \cdots, [V]_{M_c}^c}_{\text{❶: Class tokens}}, \underbrace{[V]_1^k, [V]_2^k, \cdots, [V]_{M_k}^k}_{\text{❷: Domain tokens}},
\end{aligned}
\tag{4}
$$

where $M_c$ and $M_k$ are the hyperparameter determining the number of class context tokens and domain context tokens, respectively. $\texttt{[CLASS]}$ and $\texttt{[DOMAIN]}$ is the class name and domain name, respectively, *e.g.*, the prompt is modeled as "$t = [V]_1^c, [V]_2^c, \cdots, [V]_{M_c}^c, [V]_1^k, [V]_2^k, \cdots, [V]_{M_k}^k$ [dog] [cartoon]" to match the context "a dog drawn from the cartoon domain" in PACS dataset. Note that the ❶ class tokens are shared in all classes or specific in each class, but the ❷ domain tokens are introduced to improve the vision-language embedding alignment and generalize to testing domains for improving the out-of-distribution (OOD) generalization. The reason is that the shared domain tokens can capture the domain-invariant feature and generalize to unseen testing domains, which improves OOD generalization.

We also propose a lightweight neural network named the domain meta network (DMN), which benefits from capturing both instance-specific and domain-specific information by further conditioning on the input. Specifically, let $\mathbf{V}^k = [[V]_1^k, \ldots, [V]_{M_k}^k] \in \mathbb{R}^{M_k \times B}$ denote the learnable domain-specific context vectors, and let $f_\theta : \mathbb{R}^D \to \mathbb{R}^{M_k \times B}$ be the Domain Meta Network (DMN) parameterized by $\theta$ that maps the image feature $z_v$ to an instance-conditional offset. We combine them via element-wise addition on a per-token basis:

$$
[V]_m^k(x) = [V]_m^k + \big(f_\theta(z_v)\big)_m, \qquad m = 1, \ldots, M_k. \tag{5}
$$

The resulting tokens $\{[V]_m^k(x)\}_{m=1}^{M_k}$ jointly carry domain-specific information (from $\mathbf{V}^k$) and instance-specific information (from $f_\theta(z_v)$). Note that the visual representation $z_v$ is obtained from the instance $x$, as shown in Figure 1(c). In this way, the instance-specific prompt $t_{y,j}(x)$ of class $y$ and domain $j$ is designed as:

$$
t_{y,j}(x) = \underbrace{[V]_1^c, \cdots, [V]_{M_c}^c}_{\text{❶: Class tokens}}, \quad \underbrace{[V]_1^k(x), \cdots, [V]_{M_k}^k(x)}_{\text{❷: Instance-specific domain tokens}}, y, k_j. \tag{6}
$$

**Training.** Given the instance-specific prompt $t_{y,j}(x)$ and visual representation $z_v$, the probability of assigning class labels to the image can be computed. Specifically, we model the joint posterior of class $y$ and domain $k = j$ as follows:

$$
p(y, \, k = j \mid x) = \frac{1}{K} \cdot \frac{\exp(\cos(E_t(t_{y,j}(x)), z_v)/\tau)}{\sum_{y' \in \mathcal{Y}} \exp(\cos(E_t(t_{y',j}(x)), z_v)/\tau)}. \tag{7}
$$

where the prefactor $\frac{1}{K}$ assumes a uniform prior over the $K$ training domains. The class posterior is obtained by marginalizing out the domain:

$$
p(y \mid x) = \sum_{j=1}^{K} p(y, \, k = j \mid x). \tag{8}
$$

The DMN is then trained by minimizing the following loss:

$$
\mathcal{L}_{ce} = -\frac{1}{N} \sum_{i=1}^{N} \log \left( \sum_{j=1}^{K} p(y_i, k = j \mid x_i) \right). \tag{9}
$$

Table 1: Main results of multiple training domains. For CoDoL, we report the average performance on 3 random seeds on all testing domains for each dataset. The best is bolded and the second best is underlined.

| Method | PACS | | | | | VLCS | | | | | Average |
| | Art Painting | Cartoon | Photo | Sketch | Average | Caltech | LableMe | Sun | Pascal | Average | |
|---|---|---|---|---|---|---|---|---|---|---|---|
| *ImageNet-pretrained RN50* Gulrajani & Lopez-Paz (2020) | | | | | | | | | | | |
| ERM Vapnik (1999) | 88.10 | 77.90 | 97.80 | 79.10 | 85.73 | 97.60 | 63.30 | 72.20 | 76.40 | 77.38 | 81.56 |
| IRM Arjovsky et al. (2019) | 85.00 | 77.60 | 96.70 | 78.50 | 84.45 | 97.60 | 65.00 | 72.90 | 76.90 | 78.10 | 81.28 |
| MMD Li et al. (2018b) | 84.50 | 79.70 | 97.50 | 78.10 | 84.95 | 97.10 | 63.40 | 71.40 | 74.90 | 76.70 | 80.83 |
| DANN Ganin et al. (2016) | 85.90 | 79.90 | 97.60 | 75.20 | 84.65 | 98.50 | 64.90 | 73.10 | 78.30 | 78.70 | 81.68 |
| CORAL Sun & Saenko (2016) | 87.70 | 79.20 | 97.60 | 79.40 | 85.98 | 98.80 | 64.60 | 71.70 | 75.80 | 77.73 | 81.86 |
| *CLIP-pretrained RN50* Radford et al. (2021b) | | | | | | | | | | | |
| CLIP Radford et al. (2021b) | 89.36 | 93.61 | 98.56 | 79.73 | 90.32 | 96.06 | 61.45 | 76.63 | 71.56 | 76.43 | 83.38 |
| Lin. Probing Radford et al. (2021b) | 91.29 | 90.92 | 99.02 | 85.37 | 91.65 | 98.96 | 63.37 | 79.20 | 76.39 | 79.48 | 85.57 |
| CoOp Zhou et al. (2022b) | 91.86 | 92.85 | 99.25 | 85.17 | 92.28 | 98.92 | 67.95 | 81.04 | 79.55 | 81.87 | 87.08 |
| CoCoOp Zhou et al. (2022a) | 90.89 | 92.50 | 98.21 | 84.96 | 91.64 | 99.03 | 68.50 | 81.76 | 79.90 | 82.30 | 86.97 |
| CLIP-Adapt. Gao et al. (2021) | 92.57 | 92.03 | 99.31 | 84.41 | 92.08 | 99.03 | 70.84 | 81.22 | 78.31 | 82.35 | 87.22 |
| ProGrad Zhu et al. (2022) | 92.40 | 92.65 | 99.31 | 83.70 | 92.01 | 99.07 | 70.09 | 81.22 | 78.42 | 82.23 | 87.12 |
| TPT Shu et al. (2022) | 92.70 | 93.30 | 99.07 | 83.55 | 92.16 | 99.06 | 70.87 | 81.41 | 78.22 | 82.39 | 87.28 |
| DPL Zhang et al. (2021) | 92.95 | 93.44 | 99.13 | 82.31 | 91.96 | 98.94 | 70.51 | 81.00 | 78.01 | 82.12 | 87.04 |
| StyLIP Bose et al. (2023) | 93.71 | 94.20 | 99.48 | 86.98 | 93.59 | 99.11 | 73.25 | 84.18 | 82.80 | 84.83 | 89.21 |
| **CoDoL w/o DMN** | 91.21 | 93.42 | 98.45 | 85.32 | 92.10 | 98.56 | 72.34 | 83.67 | 80.97 | 83.89 | 88.00 |
| **CoDoL w/ CMN** | 93.65 | 95.78 | 99.01 | 86.23 | 93.67 | 99.10 | 74.20 | 85.78 | 82.10 | 85.30 | 89.48 |
| **CoDoL (Ours)** | **95.31** | **96.08** | **99.50** | **87.16** | **94.51** | **99.76** | **75.87** | **86.57** | **83.81** | **86.50** | **90.51** |
| *CLIP-pretrained ViT-B/16* Radford et al. (2021b) | | | | | | | | | | | |
| CLIP Radford et al. (2021b) | 96.92 | 98.81 | 99.79 | 87.71 | 95.81 | 98.50 | 68.79 | 80.83 | 74.16 | 80.57 | 88.19 |
| Lin. Probing Radford et al. (2021b) | 97.60 | 98.95 | 99.88 | 89.73 | 96.54 | 99.21 | 68.12 | 83.62 | 79.57 | 82.63 | 89.59 |
| CoOp Zhou et al. (2022b) | 97.60 | 98.58 | 99.94 | 91.89 | 97.00 | 98.52 | 67.78 | 84.05 | 81.59 | 82.98 | 90.00 |
| CoCoOp Zhou et al. (2022a) | 97.10 | 97.98 | 99.83 | 92.00 | 96.73 | 99.22 | 72.24 | 84.79 | 78.10 | 83.59 | 90.16 |
| CLIP-Adapt. Gao et al. (2021) | 97.36 | 98.77 | 99.88 | 89.63 | 96.41 | 98.59 | 70.70 | 84.64 | 83.35 | 84.32 | 90.37 |
| ProGrad Zhu et al. (2022) | 96.80 | 98.96 | 99.59 | 90.64 | 96.50 | 99.41 | 71.30 | 83.09 | 81.50 | 83.82 | 90.16 |
| TPT Shu et al. (2022) | 96.59 | 98.35 | 99.46 | 93.50 | 96.99 | 99.20 | 71.05 | 83.20 | 81.44 | 83.72 | 91.46 |
| DPL Zhang et al. (2021) | 96.71 | 98.50 | 99.46 | 93.59 | 97.07 | 99.24 | 71.21 | 83.45 | 82.05 | 83.99 | 90.53 |
| StyLIP Bose et al. (2023) | 98.62 | 99.02 | 99.96 | 94.58 | 98.05 | 99.46 | 75.60 | 86.72 | 85.99 | 86.94 | 92.50 |
| **CoDoL w/o DMN** | 97.45 | 98.56 | 98.90 | 93.41 | 97.08 | 98.01 | 75.31 | 86.12 | 85.43 | 86.22 | 91.65 |
| **CoDoL w/ CMN** | 98.01 | 99.12 | 98.97 | 96.21 | 98.08 | 99.13 | 77.89 | 88.04 | 86.45 | 87.88 | 92.98 |
| **CoDoL (Ours)** | **98.76** | **99.76** | **99.98** | **96.56** | **98.77** | **99.78** | **78.96** | **88.62** | **86.85** | **88.55** | **93.66** |

**Inference.** Given an image $x$, we infer the image class by maximizing the posterior probability. Note that domain labels are only used at training time (to supervise which $j$ corresponds to $k_i$ in the loss); at inference time, we marginalize over all K training domains, so that no test-time domain label is required.

$$\hat{c}(x) = \arg\max_y \sum_{j=1}^{K} p(y, k = j | x). \tag{10}$$

## 4 Experiments

In this section, we evaluate the performance of CoDoL in two different OOD settings, including multiple training domains and a single training domain. Our experiments aim to answer the following problems: **Q1**: Could CoDoL achieve the robustness performance in the multiple-training-domain setting compared with state-of-the-art methods? (see Section 4.2) **Q2**: Could CoDoL have a significant performance in the challenging single-training-domain setting? (see Section 4.3). **Q3**: Does CoDoL better align the modality of vision and language? (see Section 4.4).

### 4.1 Experimental Setup

**Datasets.** We use four OOD classification benchmarks, including PACS (Li et al., 2017), VLCS (Fang et al., 2013), OfficeHome (Venkateswara et al., 2017), DigitDG (Zhou et al., 2020). (1) *PACS* is composed of four domains, which are Photo, Art Painting, Cartoon, and Sketch, with 9,990 images of 7 classes in total. (2) *VLCS* consists of four domains, which are Caltech, Labelme, Pascal, and Sun, with 6,757 images of 5 classes in total. (3) *OfficeHome* contains 13,932 images of 65 classes for classification in office and home environments, which have four domains, including Art, Clipart, Product, and Real World. (4) *DigitDG* has four different digit datasets, including MNIST (LeCun et al., 1998), MNIST-M (Ganin & Lempitsky, 2015), SVHN (Netzer et al., 2011), and SYN (Ganin & Lempitsky, 2015), which differ drastically in font style, stroke color, and background, with 19,200 images of 10 classes in total.

**Baseline methods.** We compare our proposed CoDoL with state-of-the-art methods and report experimental results based on the DomainBed (Gulrajani & Lopez-Paz, 2020) and StyLIP (Bose et al., 2023). These methods include (1) *using ImageNet pre-trained models*, *i.e.*, ERM (Vapnik, 1999), DANN (Ganin et al., 2016), CORAL (Sun & Saenko, 2016), MMD (Li et al., 2018b) and IRM (Arjovsky et al., 2019).

Table 2: Main results of multiple training domains. For CoDoL, we report the average performance on 3 random seeds on all testing domains for each dataset. The best is bolded and the second best is underlined.

| Method | OfficeHome | | | | | DigitDG | | | | | Average |
| | Art | Clipart | Real World | Product | Average | Mnist | Mnist_M | SVHN | SYN | Average. | |
|---|---|---|---|---|---|---|---|---|---|---|---|
| *ImageNet-pretrained RN50* Gulrajani & Lopez-Paz (2020) | | | | | | | | | | | |
| ERM Vapnik (1999) | 62.70 | 53.40 | 77.30 | 76.50 | 67.48 | 95.80 | 58.86 | 61.75 | 78.66 | 73.77 | 70.63 |
| IRM Arjovsky et al. (2019) | 61.80 | 52.30 | 77.20 | 75.20 | 66.63 | 95.81 | 56.78 | 64.10 | 79.45 | 74.04 | 70.34 |
| MMD Li et al. (2018b) | 63.00 | 53.70 | 78.10 | 76.10 | 67.73 | 96.52 | 58.41 | 65.32 | 78.12 | 74.59 | 71.16 |
| DANN Ganin et al. (2016) | 59.30 | 51.70 | 76.60 | 74.10 | 65.43 | 96.32 | 61.54 | 63.45 | 74.56 | 73.97 | 69.70 |
| CORAL Sun & Saenko (2016) | 64.40 | 55.30 | 77.90 | 76.70 | 68.58 | 95.23 | 61.23 | 63.74 | 76.25 | 74.11 | 71.35 |
| *CLIP-pretrained RN50* Radford et al. (2021b) | | | | | | | | | | | |
| CLIP Radford et al. (2021b) | 67.81 | 44.22 | 78.56 | 76.41 | 66.75 | 72.15 | 50.98 | 39.31 | 63.20 | 56.41 | 61.58 |
| Lin. Probing Radford et al. (2021b) | 69.54 | 49.70 | 80.06 | 81.39 | 70.17 | 78.80 | 55.22 | 43.43 | 71.41 | 62.22 | 66.20 |
| CoOp Zhou et al. (2022b) | 71.70 | 51.40 | 81.96 | 81.52 | 71.65 | 92.91 | 65.54 | 54.80 | 79.19 | 73.11 | 72.38 |
| CoCoOp Zhou et al. (2022a) | 71.55 | 51.61 | 82.25 | 82.30 | 71.93 | 93.20 | 66.47 | 57.61 | 81.03 | 74.58 | 73.26 |
| CLIP-Adapt. Gao et al. (2021) | 71.83 | 52.19 | 82.40 | 82.28 | 72.18 | 92.63 | 66.10 | 55.94 | 80.35 | 73.79 | 72.99 |
| ProGrad Zhu et al. (2022) | 71.61 | 51.80 | 82.09 | 82.00 | 71.85 | 92.75 | 66.30 | 57.50 | 81.28 | 74.45 | 73.15 |
| TPT Shu et al. (2022) | 71.83 | 52.16 | 82.40 | 81.90 | 72.07 | 93.00 | 66.54 | 57.90 | 81.29 | 74.68 | 73.38 |
| DPL Zhang et al. (2021) | 71.90 | 52.55 | 82.67 | 82.93 | 72.54 | 92.60 | 66.76 | 57.31 | 80.64 | 74.33 | 73.44 |
| StyLIP Bose et al. (2023) | 74.60 | 55.18 | 84.30 | 85.11 | 74.80 | 93.45 | 68.87 | 62.01 | 81.63 | 76.49 | 75.65 |
| CoDoL w/o DMN | 73.63 | 54.21 | 83.89 | 84.39 | 74.03 | 89.56 | 65.90 | 60.20 | 78.75 | 73.60 | 73.82 |
| CoDoL w/ CMN | 74.06 | 55.41 | 84.38 | 84.73 | 74.64 | 93.15 | 68.65 | 62.02 | 82.17 | 76.50 | 75.57 |
| **CoDoL (Ours)** | **75.85** | **57.89** | **84.73** | **87.23** | **76.43** | **94.69** | **69.83** | **63.28** | **83.75** | **77.89** | **77.16** |
| *CLIP-pretrained ViT-B/16* Radford et al. (2021b) | | | | | | | | | | | |
| CLIP Radford et al. (2021b) | 79.46 | 63.08 | 86.46 | 85.27 | 78.57 | 84.80 | 59.33 | 48.60 | 70.41 | 65.79 | 72.18 |
| Lin. Probing Radford et al. (2021b) | 81.55 | 65.70 | 87.14 | 87.32 | 80.43 | 90.42 | 62.65 | 51.70 | 75.83 | 70.15 | 75.29 |
| CoOp Zhou et al. (2022b) | 80.08 | 68.99 | 86.96 | 88.44 | 81.12 | 94.37 | 68.01 | 60.00 | 83.24 | 76.41 | 78.77 |
| CoCoOp Zhou et al. (2022a) | 79.60 | 69.35 | 86.32 | 87.51 | 80.70 | 95.55 | 70.30 | 62.59 | 85.51 | 78.49 | 79.60 |
| CLIP-Adapt. Gao et al. (2021) | 82.76 | 70.08 | 88.02 | 88.04 | 82.23 | 94.95 | 69.76 | 62.09 | 84.66 | 77.86 | 80.05 |
| ProGrad Zhu et al. (2022) | 82.57 | 70.20 | 88.60 | 88.49 | 82.46 | 94.97 | 70.21 | 63.10 | 84.77 | 78.26 | 80.36 |
| TPT Shu et al. (2022) | 82.40 | 70.63 | 88.71 | 88.05 | 82.45 | 94.67 | 71.20 | 63.50 | 84.70 | 78.51 | 80.48 |
| DPL Zhang et al. (2021) | 82.94 | 71.80 | 88.59 | 88.65 | 83.00 | 94.44 | 67.38 | 62.68 | 84.79 | 77.32 | 80.16 |
| StyLIP Bose et al. (2023) | 84.93 | 72.61 | 90.35 | 90.64 | 84.63 | 96.73 | 74.90 | 66.39 | 87.51 | 81.38 | 83.00 |
| CoDoL w/o DMN | 83.45 | 71.21 | 88.78 | 89.43 | 83.21 | 95.42 | 74.31 | 66.01 | 86.89 | 80.66 | 81.94 |
| CoDoL w/ CMN | 85.23 | 73.67 | 90.89 | 90.45 | 85.06 | 96.45 | 76.02 | 67.97 | 87.86 | 82.08 | 83.57 |
| **CoDoL (Ours)** | **85.26** | **74.65** | **91.64** | **90.98** | **85.63** | **97.02** | **76.89** | **68.32** | **88.02** | **82.56** | **84.10** |

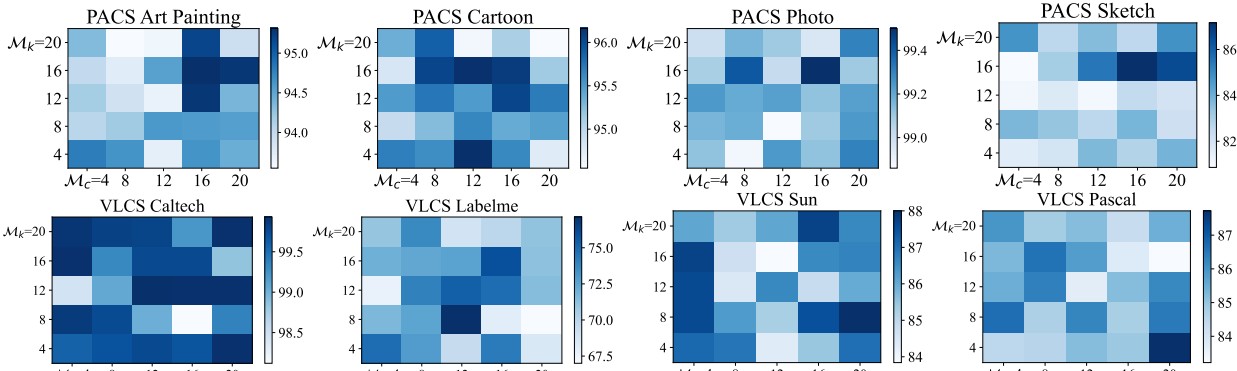

Figure 3: Ablation on the context length for $\mathcal{M}_c$ and $\mathcal{M}_k$ under the multiple-training-domain setting.

(2) *Using CLIP pre-trained models*, *i.e.*, zero-shot CLIP (Radford et al., 2021b), Linear Probing (Radford et al., 2021b), CLIP-Adapter (Gao et al., 2021), DPL (Zhang et al., 2021), CoOp (Zhou et al., 2022b), CoCoOp (Zhou et al., 2022a), ProGrad (Zhu et al., 2022), TPT (Shu et al., 2022), MIRO (Cha et al., 2022), VPT (Jia et al., 2022), CSVPT (Li et al., 2022) and StyLIP (Bose et al., 2023).

**Implementation detail.** Following StyLIP (Bose et al., 2023), in this paper, we consider two vision backbones in CLIP, *i.e.*, ResNet50 (or RN50) and ViT-B/16. We train the proposed CoDoL with a batch size of 1 for 10 epochs to ensure the model can fit into a GPU and meanwhile reduce the training time. Our proposed domain meta network (DMN) is built with a two-layer bottleneck structure, *i.e.*, Linear-ReLU-Linear, with the hidden layer reducing the input dimension by 16×. We initialize the context randomly by using different lengths. From the experimental analysis, we found that the $\mathcal{M}_c = \mathcal{M}_k = 16$ has the best performance in the multiple-training-domain setting, however, for the single-training-domain setting, the performance of $\mathcal{M}_c = \mathcal{M}_k = 8$ is best. More results can be found in Figures 3 and 4.

## 4.2 Multiple Training Domains

In this section, we evaluate CoDoL in the multiple-training-domain setting, *i.e*, there are multiple training domains and the leave-one-out strategy is used as the testing domain.

**Main Results.** In Tables 1 and 2, we report the average accuracy among all testing domains in four OOD benchmarks. We use two types of pre-trained models, including the ImageNet pre-trained model (RN50) (Gulrajani & Lopez-Paz, 2020) and the CLIP pre-trained model (RN50 and ViT-B/16) (Radford et al., 2021b). From Tables 1 and 2, we have the following findings. (1) Compared with the ImageNet pre-training model by using the supervised loss, CLIP pre-trained models achieve stunning performance on most OOD downstream tasks by using a large-scale image-language pairs to train the model based on an unsupervised contrastive loss. This indicates that the unsupervised pre-training visual-language model can alleviate the OOD problem. (2) We observe that zero-shot CLIP on DigitDG does not perform as well as supervised pre-trained methods, suggesting that textual descriptions affect the performance. However, these prompt-tuning methods, they address this problem by tuning a few parameters. For instance, zero-shot CLIP produces average target generalization accuracies of 66.75% and 76.43% on Office-Home and VLCS when used with the RN50 backbone, inferior to the performance of CoOp by at least 7%. (3) Compared with these state-of-the-art (SOTA) prompt-tuning methods, our CoDoL achieves significant performance in all OOD benchmarks, which demonstrates the effectiveness of our CoDoL in capturing the domain information in the prompt-tuning phase.

**Ablation on the domain meta network (DMN).** As shown in Tables 1 and 2, we analyze the effect of the domain meta network (DMN) by removing the DMN module (*i.e.*, CoDoL w/o DMN) or using the DMN module (*i.e.*, CoDoL). In addition, following CoCoOp (Zhou et al., 2022a), it proposes a class meta network (CMN) to generate an image conditional-input class prompt tokens, which aims to improve the generalization ability from base classes to novel classes. We also additionally introduce CMN for our proposed CoDoL, *i.e.*, CoDoL w/ CMN, which means that two lightweight neural networks are used to generate class prompt tokens and domain prompt tokens, respectively. From Tables 1 and 2, (1) Introducing the domain information can better align the image and text modalities of CLIP and improve the generalization performance of OOD tasks. CoDoL outperforms the w/o DMN in four OOD benchmarks by 3% while generalizing to unseen testing domains. (2) We also compare the performance of our CoDoL with the class meta network proposed by CoCoOP (Zhou et al., 2022a). The use of the CMN module brings little performance gain on four OOD benchmarks and even reduces the experimental results. The possible reason for this phenomenon is that the label space of training and testing domains is the same in the OOD setting, which leads to weak performance.

**Ablation on the context length.** CoDoL designs class prompt tokens $\mathcal{M}_c$ and domain prompt tokens $\mathcal{M}_k$, respectively. Following previous works (Zhou et al., 2022a;b), we consider the Cartesian product of 4, 8, 12, 16 and 20 to construct the $\mathcal{M}_c$ and $\mathcal{M}_k$. Results are shown in Figure 3 and Table 4. When $\mathcal{M}_c$ and $\mathcal{M}_k$ is 16, CoDoL achieves the best performance. An interesting phenomenon observed is that when the value of $\mathcal{M}_k$ is larger than the value of $\mathcal{M}_c$, the performance is better than the opposite case, *i.e.*, the performance of $\mathcal{M}_k = 8$ and $\mathcal{M}_c = 4$ is superior to $\mathcal{M}_c = 8$ and $\mathcal{M}_k = 4$. The phenomenon indicates that we need a longer domain context length to learn domain knowledge on OOD generalization.

### 4.3 Single Training Domains

To further evaluate the effectiveness of CoDoL, we consider a more challenging OOD setting, *i.e*, single training domain, which only uses one domain in the training phase.

**Main Results.** Training a model with a single domain poses a greater challenge compared to using multiple domains because it relies on data from only one domain to learn generalizable features. Simultaneously, the model is tested on multiple diverse targets. We focus on PACS and VLCS, and evaluate the average leave-all-but-one-domain-out performance over all possible domain combinations, as shown in Table 3. Notably, our CoDoL consistently outperforms other prompting techniques across all datasets, demonstrating an improvement of approximately 3%. As a result, we achieve new state-of-the-art results for single training domain scenarios. Specifically, in the case of VLCS, where the domains vary significantly, CoDoL showcases impressive performance across all domain combinations, surpassing the second-best method by 4%. These findings further emphasize the effectiveness of CoDoL in generating prompts that adapt well to the domain characteristics of unseen testing domains and effectively capture the visual contents.

**Ablation on the domain meta network (DMN).** The detailed experiments are shown in Table 3. We have the following findings: (1) The performance of CoDoL is the best in all settings, including datasets and

Table 3: Results of single training domain on PACS with A (Art Painting), C (Cartoon), P (Photo), and S (Sketch), and VLCS with C (Caltech), L (Labelme), P (Pascal), and S (Sun). The best is bolded.

| | | | | | | | **PACS** | | | | | | |
|---|---|---|---|---|---|---|---|---|---|---|---|---|---|
| Method | A → C | A → P | A → S | C → A | C → P | C → S | P → A | P → C | P → S | S → A | S → C | S → P | Average |
| *CLIP-pretrained RN50* Radford et al. (2021b) | | | | | | | | | | | | | |
| CoOp Zhou et al. (2022b) | 95.39 | 99.30 | 84.44 | 90.64 | 99.12 | 74.32 | 91.12 | 95.22 | 80.16 | 83.98 | 87.72 | 76.31 | 88.14 |
| CoCoOp Zhou et al. (2022a) | 94.12 | 99.28 | 82.84 | 91.88 | 98.92 | 77.13 | 80.81 | 88.72 | 76.87 | 83.35 | 85.75 | 75.27 | 86.25 |
| **CoDoL w/o DMN (Ours)** | 95.93 | 99.26 | **85.19** | 92.89 | 99.10 | **79.13** | 91.39 | 94.97 | 82.87 | **90.14** | **90.02** | 87.13 | 90.67 |
| **CoDoL w/ CMN (Ours)** | 94.43 | 99.14 | 82.90 | 92.35 | 98.44 | 73.34 | 92.04 | 94.65 | 79.57 | 88.53 | 89.78 | **92.69** | 89.82 |
| **CoDoL (Ours)** | **96.40** | **99.48** | 85.12 | **93.23** | **99.74** | 79.02 | **93.70** | **95.31** | **82.89** | 88.04 | 89.59 | 85.69 | **90.68** |
| *CLIP-pretrained ViT-B/16* Radford et al. (2021b) | | | | | | | | | | | | | |
| CoOp Zhou et al. (2022b) | 99.02 | 99.72 | 92.92 | 97.79 | 99.81 | 88.17 | 92.95 | 96.72 | 89.47 | 94.81 | 95.45 | 96.31 | 95.26 |
| CoCoOp Zhou et al. (2022a) | 98.53 | 99.64 | 93.03 | 97.51 | 99.76 | 86.16 | 93.57 | 98.43 | 90.62 | 94.66 | 94.08 | 96.59 | 95.21 |
| MAPLE Khattak et al. (2022) | 98.95 | 99.76 | 93.27 | 97.64 | 99.82 | 88.01 | 91.37 | 97.28 | 90.64 | 92.58 | 95.49 | 83.89 | 94.06 |
| VPT Jia et al. (2022) | 99.13 | 99.86 | 93.00 | 97.25 | 99.80 | 87.76 | 97.61 | 98.99 | 89.84 | 97.98 | 96.83 | 97.84 | 96.33 |
| **CoDoL w/o DMN (Ours)** | 98.70 | 99.80 | 93.19 | 97.93 | 99.84 | 88.09 | 93.78 | 98.05 | 90.58 | 92.95 | 95.73 | 85.73 | 94.04 |
| **CoDoL w/ CMN (Ours)** | 99.00 | 99.73 | 92.92 | 97.49 | 99.90 | **89.46** | 97.05 | 98.83 | **91.09** | 96.63 | **98.25** | 99.72 | 96.67 |
| **CoDoL (Ours)** | **99.39** | **99.97** | **94.86** | **98.37** | **99.86** | 89.28 | **98.68** | **99.01** | 90.88 | **98.14** | 97.19 | **99.85** | **97.12** |
| | | | | | | | **VLCS** | | | | | | |
| Method | C → L | C → P | C → S | L → C | L → P | L → S | P → C | P → L | P → S | S → C | S → L | S → P | Average |
| *CLIP-pretrained RN50* Radford et al. (2021b) | | | | | | | | | | | | | |
| CoOp Zhou et al. (2022b) | **69.97** | 84.41 | 70.25 | 90.73 | 75.91 | 70.05 | 99.25 | 60.44 | 80.66 | 93.47 | 61.19 | 83.42 | 78.31 |
| CoCoOp Zhou et al. (2022a) | 67.09 | 79.47 | 62.78 | 89.23 | 77.79 | 63.56 | 99.41 | 62.61 | 80.03 | 84.43 | 59.26 | 79.57 | 75.44 |
| **CoDoL w/o DMN (Ours)** | 68.17 | 82.89 | 65.75 | 84.51 | 75.29 | 64.53 | 99.94 | 59.56 | 81.29 | 96.93 | 58.34 | 81.70 | 76.58 |
| **CoDoL w/ CMN (Ours)** | 64.03 | 78.91 | 69.68 | 99.68 | 79.79 | 69.48 | 99.92 | 60.85 | 81.42 | 99.06 | 60.69 | 81.18 | 78.73 |
| **CoDoL (Ours)** | 68.38 | **85.69** | **70.96** | **99.84** | **81.15** | **71.88** | **99.98** | 61.61 | 81.30 | 98.35 | **61.65** | **84.80** | **80.46** |
| *CLIP-pretrained ViT-B/16* Radford et al. (2021b) | | | | | | | | | | | | | |
| CoOp Zhou et al. (2022b) | 71.35 | 87.46 | 69.95 | 94.58 | 75.12 | 69.95 | 99.76 | 58.55 | 82.95 | 95.05 | 62.11 | 84.17 | 79.25 |
| CoCoOp Zhou et al. (2022a) | 67.04 | 87.61 | 71.00 | 95.52 | 82.33 | 68.73 | 99.57 | 65.37 | 80.30 | 94.89 | 59.51 | 84.75 | 79.72 |
| MAPLE Khattak et al. (2022) | 64.53 | 83.48 | 68.32 | 98.19 | 81.74 | 67.31 | 96.70 | 63.78 | 84.97 | 95.28 | 63.45 | 85.29 | 79.42 |
| VPT Jia et al. (2022) | 66.87 | 86.67 | 70.18 | 99.61 | 85.43 | 70.26 | 98.90 | 64.07 | 85.79 | 96.47 | 63.04 | 85.26 | 81.05 |
| **CoDoL w/o DMN (Ours)** | 71.95 | 80.58 | 62.67 | 89.54 | 76.77 | 69.41 | 99.52 | 61.86 | 82.13 | **99.92** | 62.94 | 84.31 | 78.47 |
| **CoDoL w/ CMN (Ours)** | 65.05 | 79.63 | 70.29 | 99.06 | 85.16 | **71.23** | 99.83 | 66.29 | 80.78 | 99.29 | 63.15 | 82.13 | 80.16 |
| **CoDoL (Ours)** | 71.61 | **88.94** | **71.26** | **99.85** | **85.45** | 69.24 | **99.92** | **69.10** | **86.74** | 99.82 | **63.98** | **86.87** | **82.73** |

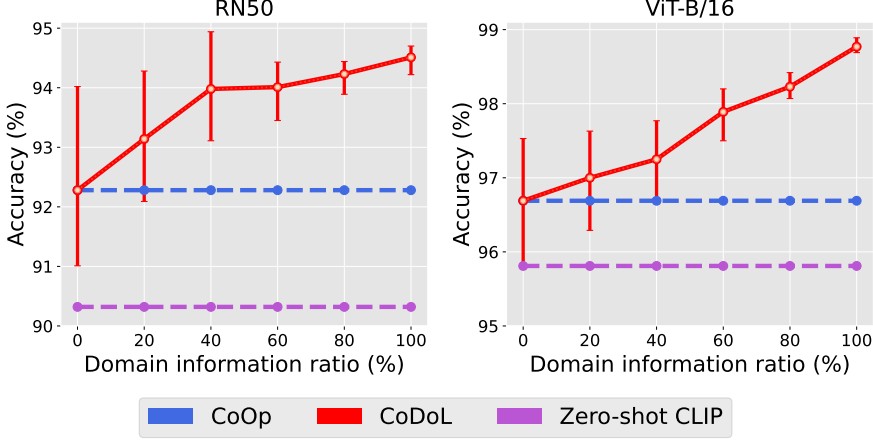

Figure 4: Ablation on the context length for $\mathcal{M}_c$ and $\mathcal{M}_k$ under the single-training-domain setting.

backbones. This indicates that CoDoL not only captures the domain-invariant information to generalize to unseen testing domains, but also combines the domain-specific representations to further align the image and text modalities and brings good performance. (2) In some training-testing-domain settings, the performance of CoDoL with CMN has an improvement by about 2% (from 71.23% to 69.24% in L→ S in VLCS under ViT-B/16), even 7% in S→P in PACS under RN50. One possible reason for this is that when there is a significant difference between two domains, even if they belong to the same class space, the model may

Figure 5: Experiments on the ratio of training domain information under multiple training domains and PACS benchmark.

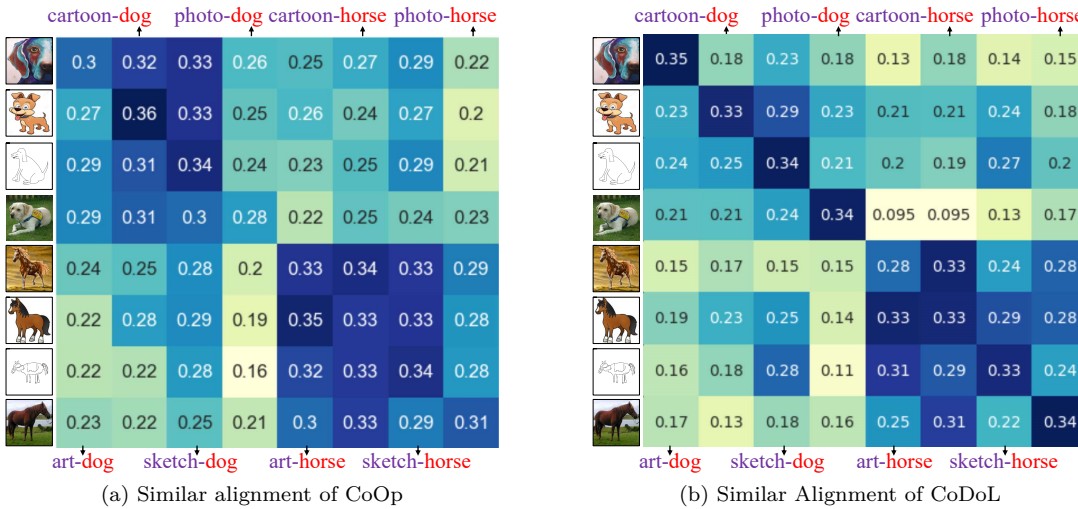

Figure 6: Experiments on vision-language alignment under multiple domains on PACS. Violet is domain and red is class.

Table 4: Ablation on different context lengths. $\mathcal{M}_c = 16$ and $\mathcal{M}_k = 16$ are used in main experimental results (*e.g.*, Tables 1 and 2 in the main paper). Best results are displayed in boldface.

| $\mathcal{M}_c$ | $\mathcal{M}_k$ | Art P. | Cartoon | PACS Photo | Sketch | Average | Caltech | LableMe | VLCS Sun | Pascal | Average | **Average** |
|---|---|---|---|---|---|---|---|---|---|---|---|---|
| | | | | | | CLIP-pretrained RN50 | Radford et al. (2021b) | | | | | |
| 4 | 4 | 94.79 | 95.67 | 99.12 | 81.59 | 92.80 | 99.58 | 74.74 | 87.09 | 84.47 | 86.47 | 89.64 |
| 4 | 8 | 94.63 | 95.58 | 98.88 | 82.00 | 92.77 | 99.69 | 72.86 | 86.88 | 84.57 | 86.00 | 89.39 |
| 4 | 12 | 93.72 | **96.16** | 99.24 | 83.71 | 93.21 | 99.76 | 69.51 | 84.35 | 85.16 | 84.70 | 88.96 |
| 4 | 16 | 94.63 | 95.61 | 99.12 | 82.82 | 93.05 | 99.67 | 74.24 | 85.46 | 84.93 | 86.08 | 89.57 |
| 4 | 20 | 94.43 | 94.75 | 99.30 | 83.86 | 93.09 | 99.92 | 68.51 | 86.98 | **87.72** | 85.78 | 89.44 |
| 8 | 4 | 94.06 | 94.95 | 99.16 | 83.76 | 92.98 | 99.85 | 71.61 | 87.59 | 86.64 | 86.42 | 89.70 |
| 8 | 8 | 94.19 | 95.25 | 99.18 | 83.35 | 92.99 | 99.74 | 72.48 | 86.27 | 84.70 | 85.80 | 89.40 |
| 8 | 12 | 94.60 | 95.63 | 98.86 | 82.61 | 92.93 | 99.02 | 77.13 | 85.26 | 86.25 | 86.92 | 89.93 |
| 8 | 16 | 94.58 | 95.38 | 99.10 | 83.83 | 93.22 | 98.12 | 68.13 | 87.49 | 84.76 | 84.63 | 88.93 |
| 8 | 20 | 94.55 | 95.46 | 99.24 | 82.23 | 92.87 | 99.36 | 67.00 | **88.00** | 86.44 | 85.20 | 89.04 |
| 12 | 4 | 94.17 | 95.49 | 99.24 | 81.08 | 92.50 | 98.46 | 67.75 | 87.59 | 85.46 | 84.82 | 88.66 |
| 12 | 8 | 93.90 | 95.72 | 99.19 | 81.70 | 92.63 | 99.07 | 73.99 | 84.45 | 86.34 | 85.96 | 89.30 |
| 12 | 12 | 93.68 | 95.49 | 99.22 | 81.03 | 92.36 | **99.93** | 75.24 | 86.58 | 83.70 | 86.36 | 89.36 |
| 12 | 16 | 95.26 | 96.03 | 99.12 | 82.46 | 93.22 | 99.92 | 74.74 | 84.85 | 85.16 | 86.17 | 89.70 |
| 12 | 20 | 94.37 | 95.69 | 99.22 | 82.03 | 92.83 | 99.92 | 71.36 | 85.97 | 86.18 | 85.86 | 89.35 |
| 16 | 4 | 94.01 | 94.82 | 99.08 | 80.86 | 92.19 | 99.92 | 71.98 | 87.70 | 85.26 | 86.22 | 89.21 |
| 16 | 8 | 93.75 | 96.03 | 99.40 | 83.09 | 93.07 | 99.31 | 72.35 | 84.75 | 86.57 | 85.75 | 89.41 |
| 16 | 12 | 94.53 | 96.15 | 99.02 | 85.46 | 93.79 | 99.76 | 72.61 | 83.84 | 85.82 | 85.51 | 89.65 |
| 16 | 16 | **95.31**±0.45 | 96.08±0.66 | **99.50**±0.08 | **87.16**±0.75 | **94.51** | 99.76±0.13 | **75.87**±0.29 | 86.57±0.37 | 83.81±0.07 | **86.50** | **90.51** |
| 16 | 20 | 95.26 | 95.14 | 99.10 | 86.51 | 94.00 | 98.85 | 71.23 | 86.68 | 83.22 | 85.00 | 89.50 |
| 20 | 4 | 94.33 | 95.35 | 99.00 | 84.74 | 93.36 | 99.89 | 71.02 | 86.07 | 85.95 | 85.73 | 89.55 |
| 20 | 8 | 93.55 | 95.86 | 99.16 | 82.61 | 92.80 | 99.81 | 73.61 | 85.36 | 84.83 | 85.9 | 89.35 |
| 20 | 12 | 93.62 | 94.60 | 99.10 | 83.60 | 92.73 | 99.79 | 68.93 | 86.07 | 85.16 | 84.99 | 88.86 |
| 20 | 16 | 95.17 | 95.09 | 98.96 | 82.56 | 92.95 | 99.21 | 69.85 | 87.70 | 84.30 | 85.27 | 89.11 |
| 20 | 20 | 93.94 | 94.55 | 99.30 | 84.78 | 93.14 | 99.92 | 71.15 | 86.58 | 85.45 | 85.78 | 89.46 |

classify them as different classes. CMN can generate conditional image features for each class, thereby enhancing the model's generalization ability in such settings.

**Ablation on the context length.** Following (Zhou et al., 2022b;a), we consider the Cartesian product of 4, 8, and 16 to construct the $\mathcal{M}_c$ and $\mathcal{M}_k$. The experiments are shown in Figure 4 and Table 5. When $\mathcal{M}_c$ and $\mathcal{M}_k$ is 8, CoDoL achieves the best performance. The differences in the single training domain are small, whereas in the multiple training domains, the models with a longer context length clearly perform better. From Figure 4 and Table 5, on the whole, when the context length is large, the performance of the model is

Table 5: Ablation on different context lengths. $\mathcal{M}_c = 8$ and $\mathcal{M}_k = 8$ are used in main experimental results (*e.g.*, Table 3 in the main paper). Best results are displayed in boldface.

**PACS**

| $\mathcal{M}_c$ | $\mathcal{M}_k$ | A→C | A→P | A→S | C→A | C→P | C→S | P→A | P→C | P→S | S→A | S→C | S→P | Average |
|---|---|---|---|---|---|---|---|---|---|---|---|---|---|---|
| | | *CLIP-pretrained RN50* Radford et al. (2021b) | | | | | | | | | | | | |
| 4 | 4 | 96.02 | 98.94 | 83.49 | 92.77 | 98.96 | 73.92 | 92.17 | 93.40 | 81.63 | 84.36 | 88.61 | 79.96 | 88.69 |
| 4 | 8 | 95.48 | 99.22 | 83.71 | 93.18 | 98.88 | 77.06 | 89.28 | 94.06 | 81.69 | 88.57 | 89.06 | 74.83 | 88.75 |
| 4 | 16 | 95.29 | 99.14 | 83.91 | 92.90 | 98.86 | **80.93** | 91.88 | 94.11 | 81.93 | 87.86 | 91.68 | 89.86 | 90.70 |
| 8 | 4 | 93.54 | 99.22 | 84.42 | 92.09 | 98.68 | 80.86 | 92.43 | 94.88 | **83.38** | 88.09 | 88.93 | 86.53 | 90.25 |
| 8 | 8 | **96.40** | **99.48** | **85.12** | **93.23** | **99.74** | 79.02 | **93.70** | **95.31** | 82.89 | 88.04 | 89.59 | 85.69 | 90.68 |
| 8 | 16 | **96.40** | 99.24 | 84.02 | **93.23** | 99.08 | 76.73 | 93.47 | 93.53 | 81.70 | 84.16 | 89.80 | 81.24 | 89.38 |
| 16 | 4 | 95.25 | 99.12 | 83.42 | 92.86 | 98.56 | 75.42 | 89.16 | 93.53 | 79.20 | 88.15 | 91.52 | 84.69 | 89.24 |
| 16 | 8 | 95.39 | 99.24 | 84.56 | 92.19 | 99.10 | 76.26 | 90.94 | 94.26 | 81.92 | **92.84** | **92.53** | 89.28 | **90.71** |
| 16 | 16 | 94.80 | 98.84 | 84.82 | 92.92 | 98.72 | 79.83 | 89.91 | 94.21 | 81.57 | 87.42 | 90.76 | **91.72** | 90.46 |
| | | *CLIP-pretrained ViT-B/16* Radford et al. (2021b) | | | | | | | | | | | | |
| 4 | 4 | 98.92 | 99.80 | **93.36** | 97.42 | 99.82 | 88.28 | 94.01 | 98.30 | 90.32 | 95.15 | 96.53 | 97.56 | 95.79 |
| 4 | 8 | 98.68 | 99.74 | 92.97 | 97.80 | 99.86 | **89.44** | 94.24 | 98.12 | 90.50 | 92.52 | **97.28** | 97.40 | 95.71 |
| 4 | 16 | 99.08 | 99.60 | 92.69 | 97.69 | 99.94 | 88.67 | 96.14 | 97.69 | 90.54 | 92.77 | 96.46 | 98.16 | 95.79 |
| 8 | 4 | 99.05 | **99.88** | 93.20 | 97.67 | 99.84 | 88.22 | 94.68 | 98.06 | 90.37 | 92.48 | 95.49 | 97.42 | 95.53 |
| 8 | 8 | 99.09 | 99.78 | 92.86 | 97.37 | 99.86 | 88.28 | 94.68 | 98.66 | 90.88 | 96.14 | 97.19 | 96.15 | 95.91 |
| 8 | 16 | 98.85 | 99.72 | 93.25 | 97.71 | **99.90** | 88.71 | 94.81 | 98.42 | **91.46** | 95.69 | 96.74 | **98.64** | **96.16** |
| 16 | 4 | 98.99 | 99.78 | 92.87 | 97.74 | 99.88 | 87.80 | 95.75 | 98.28 | 91.07 | 94.63 | 94.11 | 83.47 | 94.53 |
| 16 | 8 | 98.88 | 99.58 | 93.35 | **98.00** | 99.82 | 87.76 | 93.12 | 98.41 | 89.89 | **96.39** | 96.46 | 98.16 | 95.82 |
| 16 | 16 | **99.15** | 99.76 | 92.24 | 97.67 | 99.84 | 87.93 | **96.79** | **98.73** | 90.54 | 95.08 | 97.11 | 97.82 | 96.06 |

**VLCS**

| $\mathcal{M}_c$ | $\mathcal{M}_k$ | C→L | C→P | C→S | L→C | L→P | L→S | P→C | P→L | P→S | S→C | S→L | S→P | Average |
|---|---|---|---|---|---|---|---|---|---|---|---|---|---|---|
| | | *CLIP-pretrained RN50* Radford et al. (2021b) | | | | | | | | | | | | |
| 4 | 4 | **69.05** | 85.39 | 69.41 | 96.54 | 79.96 | 68.19 | 99.85 | 58.97 | 80.85 | 96.38 | 58.47 | 83.58 | 78.89 |
| 4 | 8 | 66.00 | 81.34 | 67.41 | 99.68 | 80.62 | 62.47 | 99.87 | 59.97 | **81.83** | 90.57 | 60.14 | 80.72 | 77.55 |
| 4 | 16 | 63.32 | 81.44 | 69.24 | 99.53 | 80.09 | 66.06 | 99.91 | 57.63 | 80.95 | 99.45 | 61.77 | 83.91 | 78.61 |
| 8 | 4 | 66.37 | 83.09 | 69.27 | 91.98 | 74.79 | 67.85 | 99.95 | 59.60 | 79.80 | 97.32 | **62.28** | 82.76 | 77.92 |
| 8 | 8 | 68.38 | **85.69** | 70.96 | **99.84** | **81.15** | 71.88 | 99.92 | **61.61** | 81.30 | **98.35** | 61.65 | **84.80** | **80.46** |
| 8 | 16 | 63.91 | 82.36 | **74.89** | 98.03 | 79.86 | **72.15** | **99.97** | 56.80 | 80.75 | 98.11 | 59.56 | 83.71 | 79.18 |
| 16 | 4 | 64.28 | 79.37 | 69.44 | 83.49 | 75.49 | 67.82 | 99.92 | 55.58 | 81.42 | 97.01 | 60.77 | 83.22 | 76.48 |
| 16 | 8 | 68.05 | 84.67 | 68.19 | 91.35 | 80.32 | 70.86 | 99.89 | 56.55 | 79.49 | 94.58 | 61.02 | 83.78 | 78.23 |
| 16 | 16 | 64.79 | 83.22 | 69.07 | 88.68 | 75.36 | 69.98 | 99.95 | 58.97 | 81.32 | 95.60 | 59.89 | 83.12 | 77.50 |
| | | *CLIP-pretrained ViT-B/16* Radford et al. (2021b) | | | | | | | | | | | | |
| 4 | 4 | 68.97 | 84.07 | 70.83 | 98.66 | 80.92 | 64.77 | 99.84 | 59.68 | 80.47 | 98.59 | 61.69 | 83.68 | 79.35 |
| 4 | 8 | 63.53 | 84.11 | 70.56 | 99.53 | 82.63 | 68.80 | 99.92 | **63.82** | 81.96 | 99.45 | 60.31 | 81.93 | 79.71 |
| 4 | 16 | 66.33 | 80.36 | 69.27 | 90.80 | 80.88 | 69.24 | 99.92 | 58.47 | 80.54 | 99.53 | 60.02 | 78.35 | 77.81 |
| 8 | 4 | 65.75 | 86.09 | 70.62 | 99.68 | 80.72 | 69.27 | **99.97** | 62.11 | 83.35 | 98.66 | 61.82 | 84.47 | 80.21 |
| 8 | 8 | **71.61** | **88.94** | 71.26 | **99.85** | **85.45** | 69.24 | 99.92 | 69.10 | **86.74** | **99.82** | **63.98** | **86.87** | **82.73** |
| 8 | 16 | 67.92 | 86.11 | **72.52** | 99.05 | 81.81 | 70.15 | 99.84 | 60.56 | 80.57 | 98.66 | 59.27 | 81.74 | 79.85 |
| 16 | 4 | 68.83 | 84.34 | 69.54 | 91.20 | 75.52 | 69.64 | 99.96 | 56.38 | 82.88 | 99.68 | 61.19 | 82.69 | 78.49 |
| 16 | 8 | 66.37 | 82.46 | 69.51 | 81.76 | 77.79 | 72.59 | 99.89 | 60.65 | 82.47 | 98.19 | 61.44 | 81.77 | 77.91 |
| 16 | 16 | 70.95 | 81.18 | 70.52 | 99.76 | 82.17 | **72.62** | 99.92 | 60.85 | 81.93 | 97.87 | 62.74 | 83.81 | 80.36 |

better than that of a small context length in the PACS and VLCS benchmarks with two backbones, which suggests that a further boost might be possible if we initialize a large context token with word embeddings. How to measure the performance gain and memory requirements resulting from an increase in context length is also a research question.

## 4.4 Quantitative Results

**Domain Information.** Although domain information is readily available in practice, we also evaluate that our CoDoL can still achieve better performance than baselines even with small amounts of domain information available. As shown in Figure 5, the result of 20% training samples with domain information outperforms the two representative baselines in both RN50 and ViT-B/16, which indicates that CoDoL is effective in OOD downstream tasks even with few domain information. An interesting finding is that the more domain information used, the more stable the inference results are than those using less domain (see the error bar in Figure 5).

Table 6: Quantitative comparison of representation quality and modality alignment on four domain-generalization benchmarks. For each dataset we report: $\Delta_{\cos} \uparrow$, $\delta_{\mod} \downarrow$, and R@1$\uparrow$. The rightmost block reports the average across all four datasets. Best results are in **bold**.

| Method | PACS | | | VLCS | | | OfficeHome | | | Digits-DG | | | Average | | |
|---|---|---|---|---|---|---|---|---|---|---|---|---|---|---|---|
| | $\Delta_{\cos}\uparrow$ | $\delta_{\mod}\downarrow$ | R@1$\uparrow$ | $\Delta_{\cos}\uparrow$ | $\delta_{\mod}\downarrow$ | R@1$\uparrow$ | $\Delta_{\cos}\uparrow$ | $\delta_{\mod}\downarrow$ | R@1$\uparrow$ | $\Delta_{\cos}\uparrow$ | $\delta_{\mod}\downarrow$ | R@1$\uparrow$ | $\Delta_{\cos}\uparrow$ | $\delta_{\mod}\downarrow$ | R@1$\uparrow$ |
| CLIP (zero-shot) | 0.21 | 0.74 | 90.2 | 0.18 | 0.81 | 82.6 | 0.17 | 0.84 | 81.5 | 0.16 | 0.89 | 65.7 | 0.18 | 0.82 | 80.0 |
| CoOp | 0.24 | 0.66 | 93.1 | 0.21 | 0.73 | 85.4 | 0.20 | 0.76 | 84.6 | 0.19 | 0.81 | 71.3 | 0.21 | 0.74 | 83.6 |
| CoCoOp | 0.25 | 0.63 | 93.9 | 0.22 | 0.70 | 86.2 | 0.21 | 0.73 | 85.5 | 0.20 | 0.78 | 72.4 | 0.22 | 0.71 | 84.5 |
| **CoDoL (Ours)** | **0.30** | **0.53** | **95.6** | **0.27** | **0.60** | **88.7** | **0.26** | **0.63** | **87.9** | **0.25** | **0.68** | **75.4** | **0.27** | **0.61** | **86.9** |

**Vision-Language Alignment.** To prove that CoDoL can better align the visual-language information than previous methods, we visualize the cosine similarity in Figure 6. CoDoL has a higher similarity than the baseline, which indicates that the domain can help align the vision and language, which also demonstrates our motivation by using the domain to improve the out-of-distribution (OOD) generalization. Moreover, we propose the lightweight neural network (DMN) for capturing both instance-specific and domain-specific information by generating input conditional tokens for images in each domain and then concatenating them with the learnable domain-specific context vectors. This indicates that CoDoL not only captures the domain-invariant information to generalize to unseen testing domains but also combines the domain-specific representations to further align the image and text modalities and bring good performance.

**Quantitative alignment across four benchmarks.** To address the concern that our alignment claim relies primarily on a qualitative visualization on PACS, we report three quantitative alignment metrics on all four benchmarks (PACS, VLCS, OfficeHome, DigitDG): (1) Cosine Similarity Gap $\Delta_{\cos}$: mean cosine similarity of matched image–text pairs minus that of mismatched pairs. Modality Gap $\delta_{\mod}$: $\ell_2$ distance between the centroid of image embeddings and that of text embeddings. Cross-modal Retrieval Recall@1 $R@1$: image-to-prompt retrieval on test domains. Across all four datasets, CoDoL consistently yields a smaller modality gap and higher retrieval Recall@1, providing quantitative evidence that the alignment improvement is not dataset-specific but holds broadly under distribution shift.

## 5 Conclusion

This paper investigates one critical issue: *how to boost alignment the vision-language modality under real-world distributional shifts*, and proposes CoDoL, a conditional domain prompt learning framework for out-of-distribution (OOD) generalization. CoDoL aims to align the image and text modalities by embedding the readily-available domain information into the prompt. We further propose a lightweight domain meta network (DMN) that generates instance-specific for images in each domain, and then concatenated with the learnable domain-specific context vectors, for capturing both instance-specific and domain-specific information. One possible limitation of this work is that the introduced domain information might lead to extra storage space and training time. Experiments on four OOD benchmark datasets have demonstrated the effectiveness of the proposed CoDoL in two OOD settings, including multiple training domains and a single training domain under two CLIP pre-trained models. In future work, we explore the scalability of the proposed method to a wider range of real-world domains beyond those considered in the current study. Investigate methods for effectively adapting the model to new and diverse domains, including methods for domain adaptation and transfer learning.

### Broader Impact Statement

This work introduces CoDoL, a method designed to enhance the robustness and generalization of vision-language models (VLMs) across diverse and unseen domains. The potential societal impacts are as follows: **Positive Impacts on System Reliability**: By improving out-of-distribution (OOD) generalization, our research contributes to the development of more reliable AI systems. In high-stakes applications such as autonomous driving or medical imaging, the ability of a model to handle data from domains not seen during

training is critical for safety and preventing catastrophic failures. **Efficiency in Adaptation**: Our proposed Domain Meta Network (DMN) is a lightweight architecture. By focusing on efficient alignment rather than massive retraining, this research supports the trend toward sustainable AI, reducing the computational energy consumption required to adapt large-scale pre-trained models to new tasks.

**Acknowledgments**

This work was supported by the Key Laboratory of Cognitive Intelligence and Content Security, Ministry of Education (Grant No.10120251107, Harbin Institute of Technology). The Key Science and Technology Program of Zhejiang Province, China (Grant No. 2026C01032). Sen Cui would like to acknowledge the financial support received from the Science and Technology Project of Beijing Municipal Science & Technology Commission (Grant No. Z251100008125030) and Shuimu Scholar program from Tsinghua University.

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
