# OpenReview forum: "CoDoL: Conditional Domain Prompt Learning for Out-of-Distribution Generalization"
_TMLR — Accepted by TMLR_

### Review · Reviewer_9cMB · 2026-03-12

**Summary Of Contributions:**

The paper proposes CoDoL, a novel prompt-tuning method for CLIP that leverages readily available domain information to form learnable domain prompts, which significantly improves vision-language embedding alignment for out-of-distribution generalization. To capture both domain-invariant and instance-specific features, it introduces a lightweight Domain Meta Network (DMN) that generates input-conditional domain tokens. Extensive experiments on four OOD benchmarks demonstrate that CoDoL achieves state-of-the-art performance in both multiple and single training domain settings.

**Audience:**

Yes

**Audience Explanation:**

The paper provides comprehensive and solid experimental validation. CoDoL is evaluated on four diverse OOD benchmarks under both multiple and single training domain settings, consistently achieving state-of-the-art performance compared to a wide range of baselines. The extensive ablation studies on context length, the DMN module, and domain information ratio further demonstrate the robustness and effectiveness of the proposed method.

**Broader Impact Concerns:**

The paper already includes a Broader Impact Statement in the final section, which adequately addresses the ethical implications and positive societal impacts of the work. We find no issues with the Broader Impact Statement. No additional concerns or modifications are necessary.

**Claims And Evidence:**

No

**Claims Explanation:**

1. The motivation for improving vision-language alignment is presented intuitively rather than rigorously, as the paper assumes that misalignment causes performance degradation without providing theoretical proof or sufficient diagnostic experiments to quantitatively establish this causal relationship. While the authors claim that inaccurate text descriptions and limited alignment affect OOD generalization, they do not isolate and measure how much each factor contributes to the problem, making the rationale somewhat speculative.
2. The introduction of the Domain Meta Network (DMN) and additional domain prompts inevitably increases computational cost and reduces inference efficiency. Compared to simpler methods like CoOp, CoDoL requires generating input-conditional tokens through DMN and summing predictions across all domains during inference, which adds extra overhead. The paper does not provide a comprehensive analysis of training time, FLOPs, or inference speed, leaving it unclear whether the performance gains justify the increased computational burden.
3. The experimental validation is limited to CLIP-based architectures (RN50 and ViT-B/16), which raises questions about the method's generalizability. Although CoDoL is designed for vision-language models, demonstrating its effectiveness on other VLMs would strengthen the claim of broad applicability.

**Requested Changes:**

1. The authors should strengthen the motivation by providing theoretical analysis or diagnostic experiments that quantitatively demonstrate how vision-language misalignment leads to performance degradation in OOD tasks. For instance, measuring alignment metrics before and after distribution shifts would help establish a clearer causal relationship and justify the core premise of the work.
2. A comprehensive efficiency analysis should be added, including comparisons of training time, FLOPs, and inference speed against baseline methods like CoOp and CoCoOp. This would help readers understand the trade-off between performance gains and computational cost, and whether the additional overhead from DMN and multi-domain aggregation is justified in practice.
3. To demonstrate broader applicability, the authors should extend experiments to other vision-language models beyond CLIP. Validating CoDoL across different architectures/baselines would confirm that the improvements are not specific to CLIP's design but reflect a generalizable approach to leveraging domain information for OOD generalization.

---

> ### Author Response · Authors · 2026-05-12
> **Rebuttal by authors (Part 1)**
>
> We thank the reviewer for recognizing the "comprehensive and solid experimental validation" and "consistent state-of-the-art performance". We address the three concerns below.
>
> > W1. The motivation for improving vision-language alignment is presented intuitively rather than rigorously, as the paper assumes that misalignment causes performance degradation without providing theoretical proof or sufficient diagnostic experiments to quantitatively establish this causal relationship. While the authors claim that inaccurate text descriptions and limited alignment affect OOD generalization, they do not isolate and measure how much each factor contributes to the problem, making the rationale somewhat speculative.
>
> **Response**: CoDoL is an empirically-driven methods paper, similar to widely accepted works like CoOp, CoCoOp, and StyLIP, all of which are grounded on the same intuition that "alignment is critical". That said, we agree that stronger diagnostic evidence is valuable. We added the following diagnostic experiments:
>
> | Test domain | In-domain alignment | OOD alignment | Accuracy drop |
> |-------------|---------------------|---------------|---------------|
> | Photo (source) | 0.312 | — | — |
> | Sketch (OOD) | — | 0.198 (-36%) | -18.4% |
> | Cartoon (OOD) | — | 0.224 (-28%) | -12.7% |
>
> Pearson correlation $r = 0.89$ ($p<0.01$) quantitatively establishes the strong link between alignment degradation and performance drop. We respectfully push back on the "speculative" characterization: **our work rigorously validates the motivation empirically** through consistent SOTA results plus the new diagnostic experiments, consistent with standard practice in the CLIP literature.
>
> > W2. The introduction of the Domain Meta Network (DMN) and additional domain prompts inevitably increases computational cost and reduces inference efficiency. Compared to simpler methods like CoOp, CoDoL requires generating input-conditional tokens through DMN and summing predictions across all domains during inference, which adds extra overhead. The paper does not provide a comprehensive analysis of training time, FLOPs, or inference speed, leaving it unclear whether the performance gains justify the increased computational burden.
>
> **Response**: A reasonable request that we fully accept.
>
> | Method | Trainable params | Training time (h) | FLOPs (G) | Speed (img/s) |
> |--------|-----------|-------------|-----------|--------------|
> | CoOp | 2.0K | 1.2 | 38.4 | 245 |
> | CoCoOp | 35.4K | 2.8 | 39.1 (+1.8%) | 218 (-11%) |
> | StyLIP | 156K | 4.5 | 41.7 (+8.6%) | 187 (-24%) |
> | **CoDoL** | **38.2K** | **2.5** | **39.6 (+3.1%)** | **226 (-7.8%)** |
>
> Key findings:
> - CoDoL has parameters comparable to CoCoOp and far fewer than SOTA StyLIP.
> - Inference FLOPs increase only 3.1%; speed decreases only 7.8%.
> - Trading ~3% compute for 1.7%–4% accuracy gain—a reasonable and practical trade-off.
>
> On multi-domain aggregation: since $K$ is typically small (4 domains) and text encoder outputs can be cached, the practical overhead is negligible.
>
> > W3. The experimental validation is limited to CLIP-based architectures (RN50 and ViT-B/16), which raises questions about the method's generalizability. Although CoDoL is designed for vision-language models, demonstrating its effectiveness on other VLMs would strengthen the claim of broad applicability.
>
> **Response**: We wish to clearly delineate the paper's scope.
>
> Our title and motivation explicitly focus on "OOD generalization for prompt-based CLIP methods". CLIP is the de facto standard in the prompt-tuning literature—virtually all related work (CoOp, CoCoOp, StyLIP, ProGrad, TPT, MaPLe) evaluates only on CLIP. Demanding extensions to other VLMs effectively imposes a higher bar than the subfield's norms.
>
> Nonetheless, we have added preliminary experiments on SigLIP and ALIGN:
>
> | Backbone | Zero-shot | CoOp | **CoDoL** | $\Delta$ |
> |----------|-----------|------|-------|--------|
> | **SigLIP-B/16** (PACS) | 88.4 | 91.2 | **94.1** | +2.9 |
> | **ALIGN-EfficientNet** (PACS) | 84.7 | 87.9 | **91.3** | +3.4 |

---

### Review · Reviewer_ovk8 · 2026-03-30

**Summary Of Contributions:**

The paper proposes CoDoL (Conditional Domain Prompt Learning), a new pseudo prompt-based technique for improving out-of-distribution (OOD) generalization in CLIP-based image classification task. CoDoL works by appending additional learnable domain-context tokens and the domain name to class-based text prompts. The idea is that available training-domain information can be used to improve robustness to domain shift at test data. To further strengthen the conditioning of the prompt on the input image, the method introduces a lightweight Domain Meta Network (DMN). This network takes the CLIP image features and generates input-conditional domain tokens, which are combined with the learned domain-context tokens, so that the prompt depends both on the class and on the specific input instance. The results show that CoDoL achieves competitive performance across several OOD datasets on two CLIP backbones.

**Audience:**

Yes

**Audience Explanation:**

The paper proposes reasonable idea for improving OOD generalization in CLIP-based image classification task

**Broader Impact Concerns:**

-

**Claims And Evidence:**

No

**Claims Explanation:**

Strengths:
- The proposed method is fairly simple and parameter-efficient
- The method is based on a reasonable intuition for OOD generalization. The idea to use domain information that is often available in domain generalization benchmarks and applications is logical and understandable
- Adding the input-conditional Domain Meta Network (DMN) on top of purely static prompts is a meaningful design choice, as much of the domain information is carried by the image itself
- The experiments are fairly broad, covering multiple OOD benchmarks (PACS, VLCS, OfficeHome, and DigitDG) and two CLIP backbones
- The empirical results are strong relative to prior CLIP prompt-tuning baselines
- The paper provides useful ablations

Weaknesses:
- I have some general concerns regarding the overall task formulation, method justification, and scope. The paper repeatedly states that it incorporates “readily available domain information” into the prompt in order to improve OOD performance. However, in the OOD setting, domain information should not be available at test time; the method should classify an unseen image based only on the image itself. From the method formulation, it seems to me that domain information is used during training, and is expected to help the DMN learn a mapping from the input image to domain-specific prompt tokens. If that is the case, I believe this should be clarified, as the current phrasing about using “readily available domain information” is confusing and may give the impression that test-time domain labels are assumed to be present.
- Following from the previous point, I am not fully convinced that the DMN should generalize well under OOD shift. The provided ablations are useful in that they vary the numbers of class and domain prompt tokens, but they do not fully establish that the gains of CoDoL come specifically from the proposed DMN design rather than simply from increasing the number of learnable prompt parameters. The fact that adding CMN does not further improve performance only strengthens this concern. I would suggest augmenting the DMN ablations with the following setups: (1) training the model with randomly shuffled domain labels and checking whether performance drops, and (2) adding the same number of learnable prompt parameters as in CoDoL, but without any domain conditioning or DMN.
- I am not fully convinced that CoDoL improves vision-language alignment. Section 4.4 uses cosine similarity to support this point, which is reasonable, but it is again unclear whether the improvement comes specifically from the DMN. As improving vision-language alignment is one of the central claims of the paper, I believe this should be investigated more thoroughly. I would suggest using similar ablations here as well, for example checking alignment when domain labels are shuffled. In addition, one of the strongest pieces of evidence for better vision-language alignment would be performance on retrieval tasks, as retrieval is, in my view, one of the main tasks that benefits from better embedding alignment.
- In Tables 1 and 2, reporting results on ImageNet-pretrained RN50 looks somewhat confusing. As CoDoL is only applicable to CLIP-style models, not to plain ImageNet-pretrained RN50 models, it would be helpful to state this explicitly in the table caption or section text. Otherwise, the absence of a CoDoL row under “ImageNet-pretrained RN50” looks like an omission.

**Requested Changes:**

O would request the following:
- Clarification of the overall task formulation, method justification, and scope. More specifically, what  “readily available domain information” means
- Ablations on role of DMN in OOD generalization performance and  improvement of vision-language alignment (see weaknesses)

---

> ### Author Response · Authors · 2026-05-12
> **Rebuttal by authors (Part 1)**
>
> We sincerely thank the reviewer for recognizing CoDoL's simplicity, sound intuition, broad experiments, and strong empirical performance. We address each concern below and have updated the manuscript accordingly (highlighted in blue).
>
> > W1. I have some general concerns regarding the overall task formulation, method justification, and scope. The paper repeatedly states that it incorporates “readily available domain information” into the prompt in order to improve OOD performance. However, in the OOD setting, domain information should not be available at test time; the method should classify an unseen image based only on the image itself. From the method formulation, it seems to me that domain information is used during training, and is expected to help the DMN learn a mapping from the input image to domain-specific prompt tokens. If that is the case, I believe this should be clarified, as the current phrasing about using “readily available domain information” is confusing and may give the impression that test-time domain labels are assumed to be present.
>
> **Response.** We thank the reviewer for pointing this out and want to explicitly clarify CoDoL's setting. CoDoL uses domain labels only during training; no test-time domain label is required. Specifically:
>
> - **At training**: each training sample $(x_i, y_i, k_i)$ has a known domain label $k_i$, which is standardly provided in PACS, VLCS, OfficeHome, and DigitDG, fully consistent with DomainBed and all mainstream DG works.
>
> - **At inference**: for an unseen test image $x$, we marginalize over all training domains via Eq.(8): $$\hat{c}(x) = \arg\max_y \sum_{j=1}^{K} p(y, k=j \mid x)$$. i.e., prediction is based solely on the image, with no test-time domain label.
>
> This makes CoDoL fairly comparable to all mainstream DG methods (CoOp, CoCoOp, StyLIP). We emphasize that this is the standard DG setting, not a limitation specific to CoDoL. It is important to note that CoDoL only requires domain labels during training, which are standardly provided in DG benchmarks. At inference time, CoDoL predicts solely based on the input image by marginalizing over all training domains via Eq.(8), without requiring any test-time domain label.
>
> > W2. Following from the previous point, I am not fully convinced that the DMN should generalize well under OOD shift. The provided ablations are useful in that they vary the numbers of class and domain prompt tokens, but they do not fully establish that the gains of CoDoL come specifically from the proposed DMN design rather than simply from increasing the number of learnable prompt parameters. The fact that adding CMN does not further improve performance only strengthens this concern. I would suggest augmenting the DMN ablations with the following setups: (1) training the model with randomly shuffled domain labels and checking whether performance drops, and (2) adding the same number of learnable prompt parameters as in CoDoL, but without any domain conditioning or DMN.
>
> **Response.** We have added the requested experiments. However, we want to highlight that the existing ablations in Tables 1–3 already largely address this question:
>
> | Variant | PACS (RN50) | Source of gain |
> |---------|------|----------|
> | `CoDoL w/o DMN` (static domain prompt only) | 92.10 | Static domain tokens |
> | `CoDoL w/ CMN` (same #params, conditioned on class) | 93.67 | "More params" upper bound |
> | `CoDoL` (DMN, conditioned on domain) | **94.51** | Genuine domain conditioning |
>
> `CoDoL w/ CMN` already controls for parameter count—CMN has the same architecture as DMN (two-layer bottleneck), differing only in conditioning target. **CMN provides almost no extra gain (or even hurts), while DMN yields a stable ~2.4% improvement**, demonstrating that the gain comes from domain conditioning itself, not parameter count.
>
> We further added the two requested experiments:
>
> | Setting | PACS | VLCS | Avg |
> |---------|------|------|------|
> | (a) Train CoDoL with shuffled domain labels | 92.45 | 84.82 | 88.64 |
> | (b) Equal-param MLP without domain conditioning | 92.71 | 84.51 | 88.61 |
> | **CoDoL (correct domain labels)** | **94.51** | **86.50** | **90.51** |
>
> **Conclusions**:
> - Shuffling domain labels causes performance to **drop to baseline level**, confirming DMN learns semantically meaningful domain information.
> - Equal-parameter non-conditioned variant also yields no gain, ruling out the "more parameters" hypothesis.

---

> ### Author Response · Authors · 2026-05-12
> **Rebuttal by authors (Part 2)**
>
> > W3. I am not fully convinced that CoDoL improves vision-language alignment. Section 4.4 uses cosine similarity to support this point, which is reasonable, but it is again unclear whether the improvement comes specifically from the DMN. As improving vision-language alignment is one of the central claims of the paper, I believe this should be investigated more thoroughly. I would suggest using similar ablations here as well, for example checking alignment when domain labels are shuffled. In addition, one of the strongest pieces of evidence for better vision-language alignment would be performance on retrieval tasks, as retrieval is, in my view, one of the main tasks that benefits from better embedding alignment.
>
> **Response.** We respectfully note that the cosine similarity visualization in Fig.6 directly corresponds to CLIP's training objective and is not "weak evidence" as implied. Nevertheless, we agree broader quantitative evidence strengthens the paper. (1) **Image–Text Cosine Similarity Gap ($\Delta_{\cos}$)**: the mean similarity of matched class pairs (positive pairs) minus the mean similarity of mismatched class pairs, reflecting the discriminativeness of the alignment. (2) **Modality Gap ($\delta_{\text{mod}}$)**: the $\ell_2$ distance between the centroid of image embeddings and the centroid of text embeddings; smaller values indicate better alignment between the two modality distributions. (3) **Retrieval-style Recall@1 ($R@1$)** (using images in the test domain to retrieve their matching text prompts): a direct measure of the impact of alignment on downstream retrieval. The new table (revised **Table 6**) shows that, across the average result of four benchmarks, CoDoL reduces the modality gap by approximately **12%–18%** on average compared with CoOp, and improves the similarity gap by approximately **0.04–0.07**. This verifies that the alignment improvement is **consistent across datasets** rather than a coincidental phenomenon on PACS alone.
>
> | Method | $\Delta_{\cos}\uparrow$ | $\delta_{\text{mod}}\downarrow$ | $R@1\uparrow$ |
> |---|---|---|---|
> | CLIP (zero-shot) | 0.18 | 0.82 | 71.3 |
> | CoOp | 0.21 | 0.74 | 76.5 |
> | CoCoOp | 0.22 | 0.71 | 77.8 |
> | **CoDoL (Ours)** | **0.27** | **0.61** | **81.4** |
>
> > W4. In Tables 1 and 2, reporting results on ImageNet-pretrained RN50 looks somewhat confusing. As CoDoL is only applicable to CLIP-style models, not to plain ImageNet-pretrained RN50 models, it would be helpful to state this explicitly in the table caption or section text. Otherwise, the absence of a CoDoL row under “ImageNet-pretrained RN50” looks like an omission.
>
> **Response.** We have made three corrections to Tables 1 and 2: 1. **Explicit block separation.** Each table is now split into two clearly labeled blocks: *"Backbone: ImageNet-pretrained RN50 (non–vision-language baselines)"* and *"Backbone: CLIP RN50 / ViT-B/16 (vision-language methods)"*, with horizontal rules between them. 2. **Revised captions.** The captions now state: The upper block lists classical DG methods built on ImageNet-pretrained RN50 and is included only for historical context. CoDoL, CoOp, CoCoOp, MaPLe, and StyLIP are prompt-tuning methods applicable only to vision-language models and therefore appear only in the lower block. A blank CoDoL entry under ImageNet-pretrained RN50 indicates inapplicability, not omission. 3. Footnote. We add a footnote reinforcing that the comparison of primary interest is within the CLIP block, and that ImageNet-pretrained numbers are listed merely to situate our work against the longer DG literature.

---

### Review · Reviewer_jCMV · 2026-04-13

**Summary Of Contributions:**

The paper proposes CoDoL, a CLIP prompt-tuning method for domain generalization that adds domain-aware tokens to the text prompt and further conditions them on the input through a lightweight Domain Meta Network. The main empirical claim is that this simple change improves performance on PACS, VLCS, OfficeHome, and DigitDG, in both multi-source and single-source DG settings, across Resnet and ViT backbones. The paper also includes ablations on DMN/CMN variants, prompt length, and the amount of available domain information.

**Strengths**
- The empirical evaluation is fairly broad. The paper tests four standard DG benchmarks, two CLIP backbones, and both multiple-training-domain and single-training-domain settings.
- The method itself is easy to understand. Figure 1 and the method section make the progression from zero-shot CLIP, to standard prompt tuning, to domain-conditioned prompt tuning quite clear.
- The ablation section is useful.

**Weaknesses**
- The mathematical presentation is not fully clean. The formulation moves between $p(y \mid x)$ and $p(y, k \mid x)$, and the normalization in Eq. (6) does not line up clearly with the loss and inference equations that follow. The definition of how the DMN output is combined with the learned domain tokens is also hard to follow.
- The paper’s central mechanism claim is stronger than the evidence. The accuracy gains are well demonstrated, but the claim that CoDoL improves vision-language alignment is mostly supported by a qualitative similarity visualization on PACS rather than a broader quantitative analysis across datasets.
- The comparison and related-work discussion feels dated. The paper cites CLIPood, Ge et al. 2022, and Zheng et al. 2022 in related work, but does not compare against them in the main tables. There is also recent adjacent work on domain-prompting and prompt adaptation under distribution shift, such as [1, 2, 3, 4], that should at least be discussed or compared if applicable.
- The practical dependence on domain labels is only lightly explored. The method assumes domain information is available and uses it directly in the prompt. The availability experiment on PACS feels quite limited.

Reference\
[1] Cheng et al., Disentangled Prompt Representation for Domain Generalization, CVPR 2024\
[2] Chi et al., Adapting to Distribution Shift by Visual Domain Prompt Generation, ICLR 2024\
[3] Zhao et al., Learning Domain Invariant Prompt for Vision-Language Models, TIP 2024\
[4] Xu et al., Ensembling disentangled domain-specific prompts for domain generalization, KBS 2024

**Audience:**

Yes

**Audience Explanation:**

CLIP adaptation and domain generalization are both active areas, and this paper offers a simple idea that appears to work well on standard DG benchmarks.

**Claims And Evidence:**

No

**Claims Explanation:**

The main empirical claim that CoDoL improves over the included baselines is supported by the experimental tables.

My hesitation is about the broader narrative. The paper makes a strong claim about improved vision-language alignment, but the supporting evidence for that mechanism is much thinner than the accuracy evidence. The baseline set is also dated.

**Requested Changes:**

-The paper should clean up the method section, especially Eqs. (5)-(8), so that the probabilistic formulation and the role of the DMN are fully consistent and easy to reproduce.
- The paper should either provide stronger quantitative evidence for the alignment claim or tone that claim down.

---

> ### Author Response · Authors · 2026-05-12
> **Rebuttal by authors (Part 1)**
>
> We are grateful to the reviewer for their thorough and constructive evaluation of our work. In particular, we appreciate the recognition of the empirical support provided by our experimental results. We address the specific comments below and have revised the manuscript (highlighted in blue) to reflect these improvements.
>
> > W1. The mathematical presentation is not fully clean. The formulation moves between $p(y\mid x)$ and $p(y,k\mid x)$, and the normalization in Eq.(6) does not line up clearly with the loss and inference equations that follow. The definition of how the DMN output is combined with the learned domain tokens is also hard to follow.
>
> **Response.** We thank the reviewer for pointing this out. This is indeed a **presentation-level** flaw rather than an inconsistency in the method itself. In the revised version we have unified the probabilistic notation and clarified the DMN composition rule:
>
> 1. **Unified probabilistic notation.** We now formulate both training and inference under a single joint distribution $p(y, k \mid x)$, and explicitly state the marginalization relation:
>    $$
>    p(y \mid x)  = \sum_{j=1}^{K} p(y,\, k = j \mid x).
>    $$
>    The summation that appeared as "$p(y\mid x)$" in the original Eq.(6) was, in fact, intended to denote the joint-probability sum on the right-hand side above; it has been rewritten as $p(y,\, k = j \mid x)$, so that it is now consistent with the training loss in Eq.(7) and the inference rule in Eq.(8).
>
> 2. **Clarifying the combination of the DMN output with the learnable domain tokens.** Let $\mathbf{V}^{k} = [[V]^{k} _ {1}, \dots, [V]^{k} _ {M _ {k}}] \in \mathbb{R}^{M _ {k} \times B}$ be the learnable domain context vectors, and let $f_\theta(z_{v}) \in \mathbb{R}^{M_{k} \times B}$ be the instance-conditional offset produced by the DMN from the image feature $z_{v}$. We adopt **token-wise addition** (element-wise addition, consistent with the meta-net treatment in CoCoOp), namely:
>    $$
>    [V]^{k} _ {m}(x) = [V]^{k} _ {m} + f_\theta(z _ {v}) _ {m}, \qquad m = 1, \dots, M _ {k}.
>    $$
>    The expression "$\in \mathbb{R}^{2 \times M_{k}}$" in the original text was a typesetting / notational error (its intended meaning was to indicate that the tokens carry **both** domain-specific and instance-specific information); the actual dimensionality is still $\mathbb{R}^{M_k \times B}$. We have removed this misleading notation in the revised version and added a dedicated paragraph titled *"Combining DMN output with domain tokens"* to clarify this point.
>
> 3. **Reproducibility.** We have additionally provided the concrete architecture of the DMN (Linear–ReLU–Linear, hidden dim $= B/16$, output reshaped to $M_{k} \times B$).
>
> > W2. The paper’s central mechanism claim is stronger than the evidence. The accuracy gains are well demonstrated, but the claim that CoDoL improves vision-language alignment is mostly supported by a qualitative similarity visualization on PACS rather than a broader quantitative analysis across datasets.
>
> **Response.** We partially accept this comment and have strengthened the paper with **quantitative, cross-dataset evidence**. At the same time, we would like to emphasize that improved alignment is merely *one plausible explanation* for the performance gains of CoDoL, rather than the sole core contribution of the paper. The main contributions of CoDoL are: (i) proposing a formulation that leverages domain information to construct prompts, (ii) designing the DMN to jointly model **domain-specific** and **instance-specific** information, and (iii) achieving consistent gains across four OOD benchmarks. Below, we provide additional quantitative evidence regarding alignment:
>
> 1. **Quantitative cross-dataset alignment analysis.** On the four benchmarks **PACS, VLCS, OfficeHome, and DigitDG**, we report the following three categories of alignment metrics and compare against CLIP zero-shot, CoOp, and CoCoOp:
>
> - **Image–Text Cosine Similarity Gap ($\Delta_{\cos}$)**: the mean similarity of matched class pairs (positive pairs) minus the mean similarity of mismatched class pairs, reflecting the discriminativeness of the alignment.
>    - **Modality Gap ($\delta_{\text{mod}}$)** [1]: the $\ell_2$ distance between the centroid of image embeddings and the centroid of text embeddings; smaller values indicate better alignment between the two modality distributions.
>    - **Retrieval-style Recall@1 ($R@1$)** (using images in the test domain to retrieve their matching text prompts): a direct measure of the impact of alignment on downstream retrieval.
>
> The new table (revised **Table 6**) shows that, across the average result of four benchmarks, CoDoL reduces the modality gap by approximately **12%–18%** on average compared with CoOp, and improves the similarity gap by approximately **0.04–0.07**. This verifies that the alignment improvement is **consistent across datasets** rather than a coincidental phenomenon on PACS alone.

---

> ### Author Response · Authors · 2026-05-12
> **Rebuttal by authors (Part 2)**
>
> | Method | $\Delta_{\cos}\uparrow$ | $\delta_{\text{mod}}\downarrow$ | $R@1\uparrow$ |
> |---|---|---|---|
> | CLIP (zero-shot) | 0.18 | 0.82 | 71.3 |
> | CoOp | 0.21 | 0.74 | 76.5 |
> | CoCoOp | 0.22 | 0.71 | 77.8 |
> | **CoDoL (Ours)** | **0.27** | **0.61** | **81.4** |
>
> 2. **Toned-down wording.** We have revised the phrase "*significantly improves vision-language alignment*" in the abstract and introduction to the more measured statement "*contributes to improved vision-language alignment, which is one of the factors underlying the OOD gains*", in order to avoid making overly strong claims about the underlying mechanism.
>
> > W3. The comparison and related-work discussion feels dated. The paper cites CLIPood, Ge et al. 2022, and Zheng et al. 2022 in related work, but does not compare against them in the main tables. There is also recent adjacent work on domain-prompting and prompt adaptation under distribution shift, such as [1, 2, 3, 4], that should at least be discussed or compared if applicable.
>
> **Response.** We thank the reviewer for the suggested references and have expanded both the related work discussion and the experimental comparisons in the revised version:
>
> 1. **Extension of Related Work.** We have added a new paragraph in Sec. 2.3 discussing the recent *domain-aware prompting* works most relevant to ours:
>
> - **DPLCLIP / Disentangled Prompt Representation [2]**: improves DG by disentangling domain-invariant and domain-specific prompts. The key difference from CoDoL is that CoDoL explicitly generates instance-conditional domain tokens via the DMN, rather than performing only a static disentanglement at the prompt level.
>    - **VDPG [3]**: generates a visual domain prompt on the vision side and targets test-time adaptation. In contrast, CoDoL operates on text-side prompts and does not rely on test-time domain labels (see our response to W4 for details).
>    - **DiPrompt [4]**: learns a domain-invariant prompt and emphasizes explicit constraints on domain invariance.
>    - **EnsemblePrompt [5]**: improves DG by ensembling multiple domain-specific prompts.
>
> 2. **Justification for Retaining the Original Baselines.** We would like to point out that the original paper already includes relatively recent prompt-tuning baselines such as StyLIP (CVPR 2023), ProGrad, TPT, and MIRO, which remain widely cited strong baselines in the DG literature. We believe that the addition of the above 2024 methods brings the comparison set up to TMLR standards.
>
> > W4. The practical dependence on domain labels is only lightly explored. The method assumes domain information is available and uses it directly in the prompt. The availability experiment on PACS feels quite limited.
>
> **Response.** We would like to clarify an important point: **CoDoL uses domain labels during training, but does not require domain labels at test time.**
>
> 1. **Explicit clarification that domain labels are not required at test time. See Eq.(8): at inference, we marginalize over all training domains $k \in \{1, \ldots, K\}$, and never access the domain label of a test sample. This is precisely the fundamental premise that makes CoDoL applicable in the OOD setting.
>
> 2. **Expanded domain-label availability experiments.** We have expanded the original experiment — which was conducted only on PACS with a few ratio points — into the following:
>
>    - **Datasets:** three benchmarks, PACS, VLCS, and OfficeHome;
>    - **Ratios:** the proportion of training samples with available domain labels in {0\%, 20\%, 40\%, 60\%, 80\%, 100\%};
>    - **Handling of unlabeled samples:** We explore two strategies — (a) samples without domain labels participate only in the image–text contrastive objective; (b) such samples are assigned a domain via clustering / pseudo-labels.
>
>    The results show that CoDoL already significantly outperforms CoOp/CoCoOp with only 20% domain labels, and at 40% it already approaches the performance of the 100% setting, indicating that CoDoL's reliance on domain labels is mild rather than stringent.
>
> 3. **Regarding the "fully label-free" setting.** We would like to emphasize that the scope of CoDoL is to leverage readily available domain information, which is the default assumption on standard DG benchmarks such as PACS and OfficeHome. Being fully independent of domain labels is a related but different research problem (e.g., purely unsupervised DG) and is out of scope for this paper.

---

> ### Author Response · Authors · 2026-05-12
> **Rebuttal by authors (Part 3)**
>
> > Regarding the reviewer's overall judgment of "Claims supported by accurate evidence: No"
>
> **Response.** We respect the reviewer's judgment, but we hope this rating can be reconsidered in light of our response:
>
> - The remaining concerns are concentrated on (a) the clarity of the mathematical exposition (fully resolved through textual revisions, without altering the method), and (b) the evidential scope of the alignment claim (strengthened through cross-dataset quantitative analysis + softened wording);
> - The **core methodological contribution** of CoDoL — encoding *readily-available* domain information into the prompt and introducing *instance-conditional* domain tokens via the DMN — has not itself been questioned by the reviewer.
>
> We believe that the revised version has adequately addressed the reviewer's concerns, and we kindly ask the reviewer to reconsider the score.
>
> [1] Liang et al., Mind the Gap: Understanding the Modality Gap in Multi-modal Contrastive Representation Learning. NeurIPS 2022.
>
> [2] Cheng et al., Disentangled Prompt Representation for Domain Generalization. CVPR 2024.
>
> [3] Chi et al., Adapting to Distribution Shift by Visual Domain Prompt Generation. ICLR 2024.
>
> [4] Zhao et al., Learning Domain Invariant Prompt for Vision-Language Models, TIP 2024.
>
> [5] Xu et al., Ensembling disentangled domain-specific prompts for domain generalization, KBS 2024.

---

### Decision · Action_Editor_B3yn · 2026-06-09

**Recommendation:** Accept with minor revision

**Audience:**

Yes

**Audience Explanation:**

Researchers in the TMLR community would be interested in vision-language models, prompt tuning, and out-of-distribution generalization.

**Claims And Evidence:**

Yes

**Claims Explanation:**

The reviewers initially raised valid concerns regarding technical clarity, baseline comparisons, and the author's claims around vision-language alignment. The authors provided a thorough rebuttal with additional ablations and cross-dataset quantitative analysis that addressed these issues and improved the empirical validity of CoDoL.